# Ret function in muscle stem cells points to tyrosine kinase inhibitor therapy for facioscapulohumeral muscular dystrophy

Louise A Moyle[1,2], Eric Blanc[1,3,4], Oihane Jaka[5], Johanna Prueller[1], Christopher RS Banerji[1], Francesco Saverio Tedesco[2], Stephen DR Harridge[5], Robert D Knight[6]*, Peter S Zammit[1]*

[1]Randall Division of Cell and Molecular Biophysics, King's College London, London, United Kingdom; [2]Department of Cell and Developmental Biology, University College London, London, United Kingdom; [3]Core Unit Bioinformatics, Berlin Institute of Health, Berlin, Germany; [4]Institute of Pathology, Charite Universitatsmedizin Berlin, Berlin, Germany; [5]Centre of Human and Aerospace Physiological Sciences, King's College London, London, United Kingdom; [6]Craniofacial Development and Stem Cell Biology, King's College London, London, United Kingdom

**Abstract** Facioscapulohumeral muscular dystrophy (FSHD) involves sporadic expression of DUX4, which inhibits myogenesis and is pro-apoptotic. To identify target genes, we over-expressed DUX4 in myoblasts and found that the receptor tyrosine kinase *Ret* was significantly up-regulated, suggesting a role in FSHD. *RET* is dynamically expressed during myogenic progression in mouse and human myoblasts. Constitutive expression of either *RET9* or *RET51* increased myoblast proliferation, whereas siRNA-mediated knockdown of *Ret* induced myogenic differentiation. Suppressing RET activity using Sunitinib, a clinically-approved tyrosine kinase inhibitor, rescued differentiation in both DUX4-expressing murine myoblasts and in FSHD patient-derived myoblasts. Importantly, Sunitinib also increased engraftment and differentiation of FSHD myoblasts in regenerating mouse muscle. Thus, DUX4-mediated activation of *Ret* prevents myogenic differentiation and could contribute to FSHD pathology by preventing satellite cell-mediated repair. Rescue of DUX4-induced pathology by Sunitinib highlights the therapeutic potential of tyrosine kinase inhibitors for treatment of FSHD.

*For correspondence: robert. knight@kcl.ac.uk (RDK); peter. zammit@kcl.ac.uk (PSZ)

**Competing interests:** The authors declare that no competing interests exist.

## Introduction

Facioscapulohumeral muscular dystrophy (FSHD) is a prevalent myopathy, affecting 12/100,000 people (*Deenen et al., 2014*). Although clinical severity is highly variable, skeletal muscle weakness often appears first in the facial and shoulder girdle muscles, with muscles of the trunk and lower extremities becoming affected as the disease progresses. Distinctive features of FSHD include asymmetric skeletal muscle wasting and scapular winging (*Tawil, 2008*). Analysis of myoblasts from FSHD patients has revealed direct functional impairments, including increased susceptibility to oxidative stress (*Winokur et al., 2003a, 2003b; Celegato et al., 2006; Macaione et al., 2007; Barro et al., 2010*), up-regulation of apoptotic markers (*Winokur et al., 2003a, 2003b; Sandri et al., 2001; Vanderplanck et al., 2011; Laoudj-Chenivesse et al., 2005*) and repression of *MYOD* and MYOD-target genes (*Winokur et al., 2003b; Celegato et al., 2006*). Additionally, myogenic differentiation results in myotubes with either an atrophic, or hypertrophic and highly disorganised morphology (*Barro et al., 2010; Vanderplanck et al., 2011; Tassin et al., 2012; Ansseau et al., 2009*). Defects

in myoblast function are likely to result in impaired muscle maintenance, directly contributing to clinical symptoms (*Morgan and Zammit, 2010*).

FSHD is divided into two clinically indistinguishable disorders. FSHD1 (1A) (OMIM #158900) is associated with the contraction of a macrosatellite repeat named D4Z4 in the subtelomeric region 4q35 (*Tawil, 2008*; *van Deutekom et al., 1993*). Embedded within each 3.3 kb D4Z4 repeat unit is an open reading frame (ORF) for *Double homeodomain protein 4 (DUX4)*, an intron-less retrogene (*Dixit et al., 2007*; *Gabriëls et al., 1999*). Contraction of D4Z4 units to between 1–10 is associated with de-repression of chromatin (*van Deutekom et al., 1993*; *Zeng et al., 2009*; *de Greef et al., 2009*). If this occurs on a permissive chromosomal 4qA haplotype containing a polyadenylation site within the flanking pLAM region, *DUX4* mRNA transcript from the distal D4Z4 repeat is stabilised and DUX4 protein translated (*Dixit et al., 2007*; *Lemmers et al., 2010*; *Snider et al., 2010*). In the rarer FSHD2 (1B) (OMIM #158901), de-repression of the D4Z4 macrosatellite occurs despite the presence of >11 D4Z4 repeats, but still requires the permissive 4qA haplotype containing the polyadenylation signal (*de Greef et al., 2009*, *2010*). FSHD2 has recently been linked to mutations in the *Structural maintenance of chromosomes flexible hinge domain containing* 1 (*SMCHD1*) gene, encoding a protein involved in DNA methylation (*Lemmers et al., 2012*). Recently, mutations in *DNA methyltransferase 3B (DNMT3B)* have also been implicated in perturbed epigenetic regulation in FSHD (*van den Boogaard et al., 2016*). Taken together, the consensus is that aberrant DUX4 expression underlies pathogenesis in both FSHD1 and FSHD2 (*Tawil et al., 2014*).

Two main *DUX4* mRNA transcripts can be derived from the ORF in a D4Z4 repeat: *DUX4-fl* (full length) mainly expressed in germline cells, and the alternatively spliced *DUX4-s* (short) transcript expressed in some somatic cells. FSHD skeletal muscle expresses *DUX4-fl* whereas this transcript is not usually found in healthy control muscle (*Snider et al., 2010*), although very low levels have been reported in some studies (*Jones et al., 2012*; *Tassin et al., 2013*). Expression of DUX4 in myoblasts recapitulates the pathogenic phenotype of myoblasts from FSHD patients (*Vanderplanck et al., 2011*; *Yao et al., 2014*; *Mitsuhashi et al., 2013*; *Bosnakovski et al., 2008a*; *Knopp, 2011*; *Geng et al., 2012*; *Bosnakovski et al., 2014*; *Kowaljow et al., 2007*; *Wallace et al., 2011*; *Geng et al., 2011*; *Banerji et al., 2015*). Indeed, suppression of *DUX4* expression using siRNA or anti-sense oligonucleotides rescues the atrophic phenotype of FSHD myotubes in vitro (*Vanderplanck et al., 2011*). DUX4 is a potent transcription factor and analysis of DUX4-expressing myoblasts has revealed that major transcriptional pathways such as cell cycle regulation, glutathione redox metabolism, myogenic differentiation and Wnt signalling are disrupted (*Bosnakovski et al., 2008a*; *Geng et al., 2012*; *Banerji et al., 2015*). In addition, many germline and neural genes are upregulated (*Banerji et al., 2015*; *Dandapat et al., 2013*). However, DUX4 is at very low levels in FSHD biopsies (*Tassin et al., 2013*). Despite this, we, and others, have shown that the transcriptional landscape of genes altered by DUX4 is much more prominent in FSHD patient-derived material (*Yao et al., 2014*; *Banerji et al., 2015*). Therefore, identification of DUX4-induced pathways that contribute to pathology in FSHD and are targetable with drugs provides a useful route for potential FSHD-specific therapies. We have focussed on the effects of DUX4 on myoblasts, to identify approaches to improve muscle repair and regeneration.

At present, there is no single mammalian animal model that encompasses the genetic and pathophysiological spectrum of FSHD (*Lek et al., 2015*). To model FSHD, we have used retroviral-mediated expression of DUX4, in conjunction with constitutively active and dominant-negative versions, in primary murine satellite cells cultured ex vivo. Comparing gene expression data from DUX4-expressing murine satellite cell-derived myoblasts to a meta-analysis of published microarrays from FSHD patient biopsies and primary cultures (*Banerji et al., 2015*), we revealed a significant overlap between DUX4-expressing murine satellite cell-derived myoblasts and human FSHD muscle (*Knopp et al., 2016*).

We found that the receptor tyrosine kinase (RTK) *Rearranged during transfection (RET)* is up-regulated in DUX4-expressing myoblasts (*Banerji et al., 2015*; *Dandapat et al., 2013*). *RET* is a transmembrane RTK containing 4 cadherin-like repeats, a cysteine-rich area, a calcium-binding site on the extracellular region and several intracellular tyrosine kinase domains (*Santoro et al., 2004*; *Airaksinen and Saarma, 2002*). There are two main *RET* isoforms; *RET9* and *RET51*, alternatively spliced in the 3′ region, resulting in an additional tyrosine at Y1096 on RET51 (*Jain et al., 2006*). These two main *RET* isoforms are differentially expressed, regulated and have diverse roles in

development and tissue homeostasis (*Yoong et al., 2005*; *Tsui-Pierchala et al., 2002*; *Richardson et al., 2012*; *Jain et al., 2010*).

RET is activated by the secreted Glial cell-derived neurotrophic factor (GDNF) family ligands (GFL): GDNF, neurturin (NRTN), artemin (ARTN) and persephin (PSPN). GFLs bind with high affinity to one of four glycosylphosphatidylinositol (GPI)-anchored co-receptors of the GDNF Family Receptor (GFRα) family. GFL:GFRα complexes then recruit two RET molecules, triggering RET dimerisation and trans-phosphorylation of the intracellular region of both molecules. This leads to recruitment of adaptor proteins and activation of downstream signalling cascades such as MAPK, PI3/AKT, JAK/STAT, JNK, ERK5, SRC and PLCγ (*Santoro et al., 2004*; *Airaksinen and Saarma, 2002*; *Runeberg-Roos and Saarma, 2007*; *Mulligan, 2014*). RET9 and RET51 activate PI3K/AKT and ERK signalling with different dynamics and have distinct abilities to recruit Src or Frs2, suggesting differential activation of downstream targets by the two main RET isoforms (*Tsui-Pierchala et al., 2002*; *Ishiguro et al., 1999*; *Besset et al., 2000*).

Coordinated RET signalling is crucial for development of multiple tissues, including many neuronal cell types, kidneys and the thyroid gland (*Runeberg-Roos and Saarma, 2007*; *Arighi et al., 2005*). Indeed, RET knockout mice can be embryonic lethal with a wide range of organ defects (*Jain et al., 2010*; *Schuchardt et al., 1994*; *de Graaff et al., 2001*). RET signalling has also been shown to regulate spermatogonial and haematopoietic stem cell function (*Naughton et al., 2006*; *Fonseca-Pereira et al., 2014*). Dysregulated RET signalling is associated with disease: inactivating mutations of *RET* lead to a form of neurocristopathy called Hirschsprung's disease (*Amiel et al., 2008*), whereas mutations of *RET* resulting in constitutively active isoforms are associated with several forms of inherited and somatic cancers (*Arighi et al., 2005*; *Borrello et al., 2013*). For example, mutations at codon 634 of *RET* are found in approximately 84% of multiple endocrine neoplasia (MEN2A) patients (*Santoro et al., 2004*; *de Graaff et al., 2001*; *Mulligan et al., 1993*).

Studies detailing expression of *Ret*, the GFL's and GFRα's in developing or adult rat, mouse and human skeletal muscle report expression of at least some RET signalling components, predominantly during development e.g. (*Worby et al., 1998*; *Naveilhan et al., 1998*; *Russell et al., 2000*; *Lindahl et al., 2001*; *Yang et al., 2006*; *Mikaels et al., 2000*; *Golden et al., 1999*; *Yang and Nelson, 2004*). Interestingly, *Ret* and *Gdnf* mRNA transcripts have been shown in mesenchymal cells of the second branchial arch in E9-9.5 mouse embryos (*Golden et al., 1999*; *Natarajan et al., 2002*), from which some facial muscles develop. We recently reported that *ret* is important for skeletal muscle development in zebrafish, where ret-gfrα3-artemin signalling is required to regulate myogenic differentiation in a subset of muscle precursors deriving from the first and second brachial arches (*Knight et al., 2011*).

Here we show that Ret is a novel mediator of satellite cell function, contributing to the regulation of proliferation and initiation of myogenic differentiation. Critically, DUX4-mediated Ret signalling is pathogenic to satellite cells, contributing to impaired differentiation. However, blocking DUX4-mediated Ret signalling using siRNA or the RTK inhibitor Sunitinib, leads to an increase in the myogenic capacity of DUX4-expressing murine satellite cells. Importantly, Sunitinib also improves proliferation and differentiation in FSHD patient-derived myoblasts both *in vitro*, and *in vivo* after grafting into regenerating muscle in mice. These observations highlight using RTK inhibitors as a potential therapeutic strategy for FSHD.

## Results

### Ret and Ret co-receptors are expressed in murine myoblasts

To determine whether *Ret* is expressed in murine satellite cells, myofibres with their associated satellite cells were isolated and either fixed immediately (T0), or cultured for 24 (T24) or 72 (T72) hours. Co-immunolabelling to detect Pax7 and Ret51 of fixed myofibres at T0 and T24 revealed that Ret51 was barely detectable in quiescent satellite cells, but expression increased after 24 hr of activation (*Figure 1A and B*). This correlated with an increase in phospho-Ret Tyr1062 immunolabelling, indicative of active Ret signalling. By T72, a proportion of satellite cell-derived myoblasts stop expressing Pax7 and commit to myogenic differentiation, expressing Myogenin (*Zammit et al., 2004*). Ret51 was found in both Myogenin-negative uncommitted myoblasts and Myogenin-positive differentiating myoblasts (*Figure 1C*).

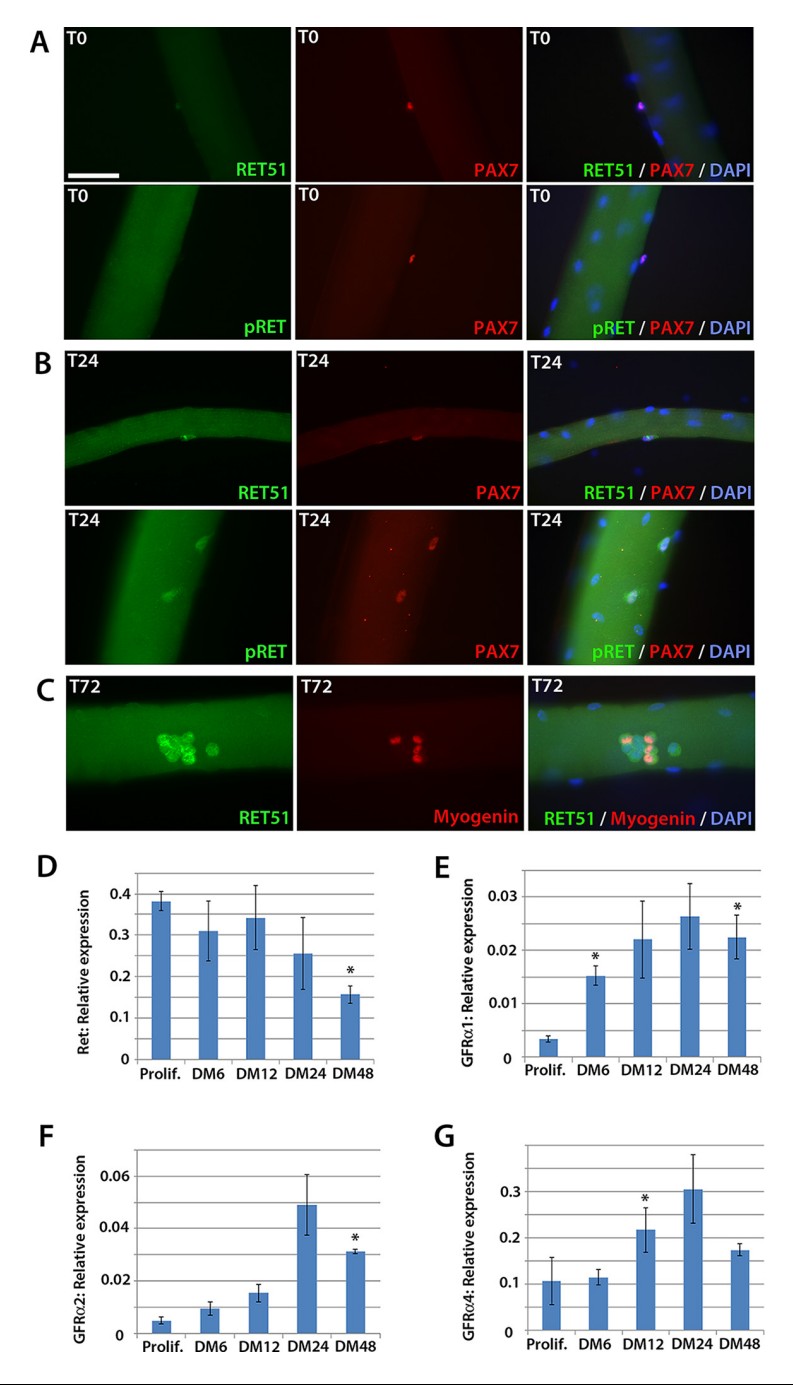

**Figure 1.** Expression dynamics of *Ret* and *Gfrα* co-receptors during myogenic progression in murine satellite cell-derived myoblasts. Co-immunolabelling of Ret in satellite cells associated with isolated myofibres. (**A**–**C**) Expression of Ret51 and phosphorylated (Y1062) Ret (pRet) in quiescent (T0) Pax7-positive satellite cells (**A**), Pax7-positive satellite cells cultured for 24 hr (T24) (**B**) and expression of Ret51 in differentiation-committed Myogenin-positive and Myogenin-negative satellite cells (**C**). DAPI (blue) was used as a nuclear counterstain. (**D**–**G**) Relative expression dynamics of *Ret* and its co-receptors *Gfrα1, Gfrα2* and *Gfrα4* during satellite cell proliferation (prolif.) and through differentiation (DM 6–48 hr), normalised to *Tbp*. Data is mean ± SEM from 3 independent experiments using 3 mice. An asterisk denotes significant difference (p<0.05) from the expression level in proliferating cells, as determined using a paired Student's *t*-test. Scale bar equals 50 μm.

Next, the expression of *Ret* and its *Gfrα* co-receptors was profiled during myogenic progression. Satellite cell-derived myoblasts were isolated from three 8-week old C57BL/10 mice and mRNA harvested from plated expanded cultures in proliferation medium or after 6, 12, 24 or 48 hr in differentiation medium (*Figure 1D–G*). *Ret, Gfrα1, Gfrα2* and *Gfrα4,* but not *Gfrα3*, were robustly expressed in these myoblast cultures. Expression of *Ret* decreased during myoblast differentiation, whereas *Gfrα1, Gfrα2* and *Gfrα4* increased upon differentiation.

## RET is expressed in human myoblasts

We next investigated expression of *RET* in human primary satellite cell-derived myoblasts (*Figure 2*). Primary myoblasts were isolated from vastus lateralis of three 19-year old females and the CD56+ fraction purified using magnetic-activated cell sorting (*Agley et al., 2015*). CD56+ satellite cell-derived myoblasts from each individual were cultured in proliferation medium or in differentiation medium for 1, 2, 3 or 4 days. Cultures were fixed and immunolabelled to confirm differentiation (*Figure 2A*). RNA was extracted from sister cultures for RT-qPCR analysis, which revealed that *RET* was detectable at low levels in proliferating and differentiating human primary myoblasts (*Figure 2B*), with control genes *CYCLIN D1, MYOD, MYOG* and *MYHC* used to confirm proliferation/differentiation status (*Figure 2C–F*).

## Ret is required for satellite cell proliferation

To test whether satellite cells require *Ret* for normal function, we performed siRNA-mediated knockdown. Expanded murine satellite cell cultures were transfected with 20 nM of *Ret* siRNA directed at both *Ret9* and *Ret51* isoforms, or a scrambled-sequence control siRNA, for 48 hr. Knockdown was assessed by RT-qPCR, with a mean knockdown efficiency of ~80% achieved. siRNA-treated satellite cell-derived myoblasts were pulsed for two hours with the thymidine analogue 5-Ethynyl-2′-deoxyuridine (EdU), which incorporates into DNA during S phase of the cell cycle and can be used to measure relative proliferation rate, and then immunolabelled for Pax7 (*Figure 3A*). Knockdown of *Ret* significantly reduced the proportion of myoblasts that had incorporated EdU, from a mean ± SEM of 44.0 ± 4.5% to 28.0 ± 3.2% (*Figure 3B*). Sister cultures of siRNA-treated myoblasts were then co-immunostained with phosphorylated-histone H1 and phosphorylated-histone H3 protein to identify specific stages of the cell cycle (*Lu et al., 1994*; *Knopp et al., 2013*; *Hendzel et al., 1997*). This confirmed that *Ret* knockdown led to decreased proliferation, with significantly fewer cells in S and G1 phases (*Figure 3C*).

SiRNA-mediated knockdown of *Ret* was also associated with a small decrease in the proportion of myoblasts containing Pax7 protein (*Figure 3D*). The reduced number of proliferating, Pax7 expressing myoblasts when *Ret* is knocked down suggests that *Ret* contributes to maintaining myoblasts in an undifferentiated state. We also observed a reduction of *Pax7* expression in response to *Ret* knockdown (*Figure 3E*). Although *Pax7* is genetically upstream of *MyoD* and *Myf5* (*Relaix et al., 2005*), *Ret* knockdown did not alter *MyoD* (*Figure 3F*) or *Myf5* expression (*Figure 3G*).

## Knockdown of Ret enhances myogenic differentiation of satellite cell-derived myoblasts

Satellite cell-derived myoblasts treated with *Ret* siRNA had reduced *Pax7* expression and proliferation rate, implying that they may be differentiating prematurely. To investigate this, cultures were treated with control or *Ret* siRNA, plated at high density and incubated in differentiation medium for 48 hr, before immunolabelling for Myogenin and Myosin Heavy Chain (MyHC) (*Figure 3H and I*). The proportion of Myogenin-positive nuclei was significantly increased in *Ret* siRNA-treated cultures relative to control siRNA-treated cells (85.4 ± 4.0% versus 69.1 ± 4.0%) (*Figure 3J*). In contrast, the proportion of nuclei in MyHC-expressing multinucleate myotubes (presented as the fusion index: the proportion of nuclei in myotubes of 2 or more nuclei, *Figure 3K*) was unaltered in control cells (60.0 ± 3.0%) compared to cells treated with *Ret* siRNA (62.7 ± 2.2%). To determine if the extent of myoblast fusion was affected, myotubes were classified by the number of nuclei per myotube (small (2>4), medium (5>9), large (10>24) and extra-large (25+). *Ret* siRNA knockdown led to more large myotubes (10.7 ± 1.9%) relative to control (5.4 ± 1.8%, *Figure 3L*).

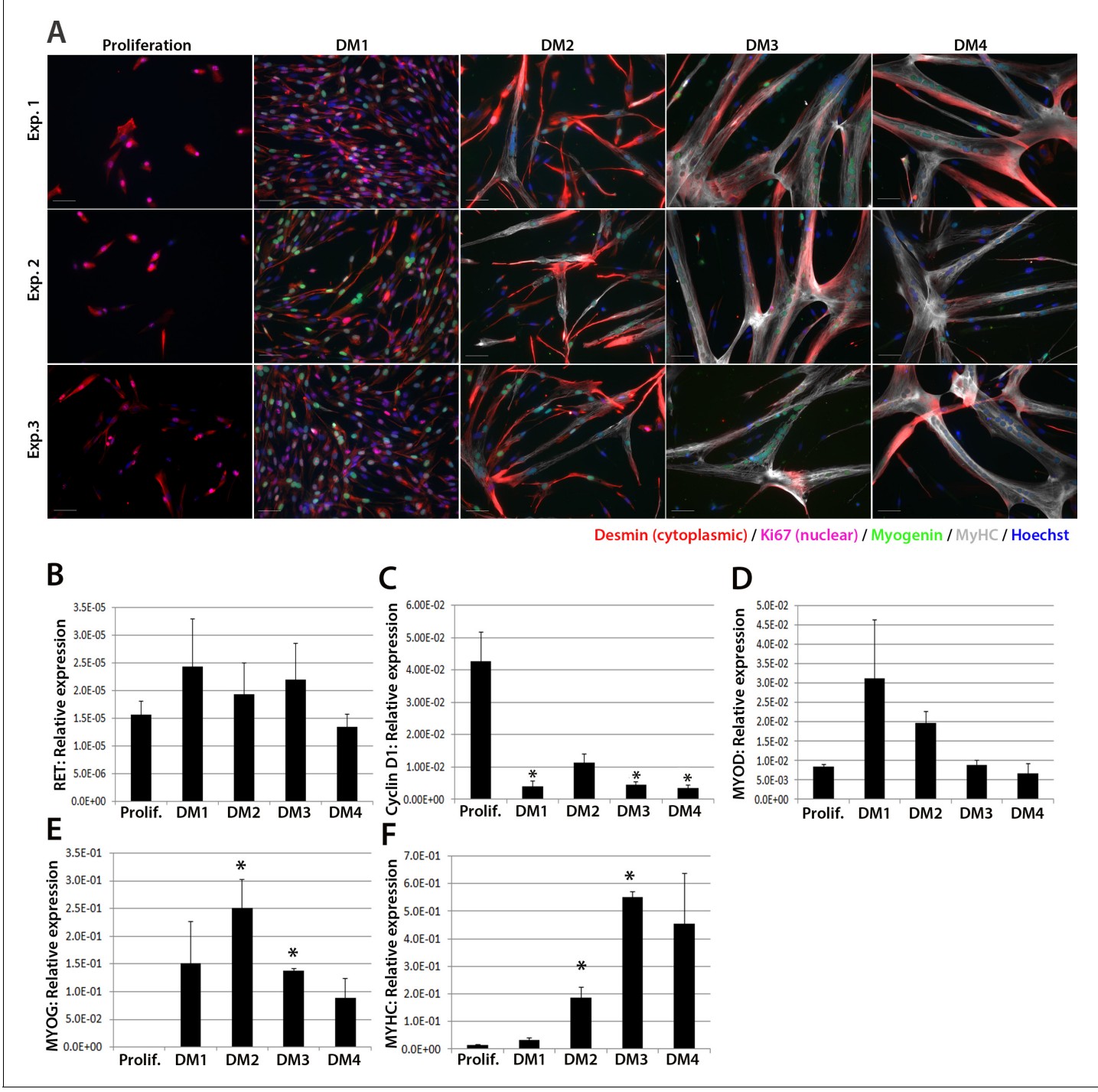

**Figure 2.** RET is expressed in proliferating and differentiating primary human satellite cell-derived myoblasts. (A) Immunolabelling of primary human myoblasts extracted from the *vastus lateralis* of 3 individuals (Exp. 1 to 3) co-immunolabelled for Desmin (red – cytoplasmic), Ki67 (mauve – nuclear), Myogenin (green – nuclear), MyHC (grey – cytoplasmic) and counterstained with DAPI (blue – nuclear). (B–F) Relative expression of (B) *RET*, (C) *Cyclin D1*, (D) *MYOD*, (E) *MYOG* and (F) *MYHC* transcription during proliferation (Prolif.) and after 1, 2, 3 and 4 days in differentiation medium (DM1-4), measured by RT-qPCR and normalised to *RPLPO* housekeeping gene. Data is mean ± SEM where an asterisk denotes significant difference (p<0.05) from the expression level in proliferating cells, as determined using a paired Student's *t*-test. Scale bar equals 50 µm.

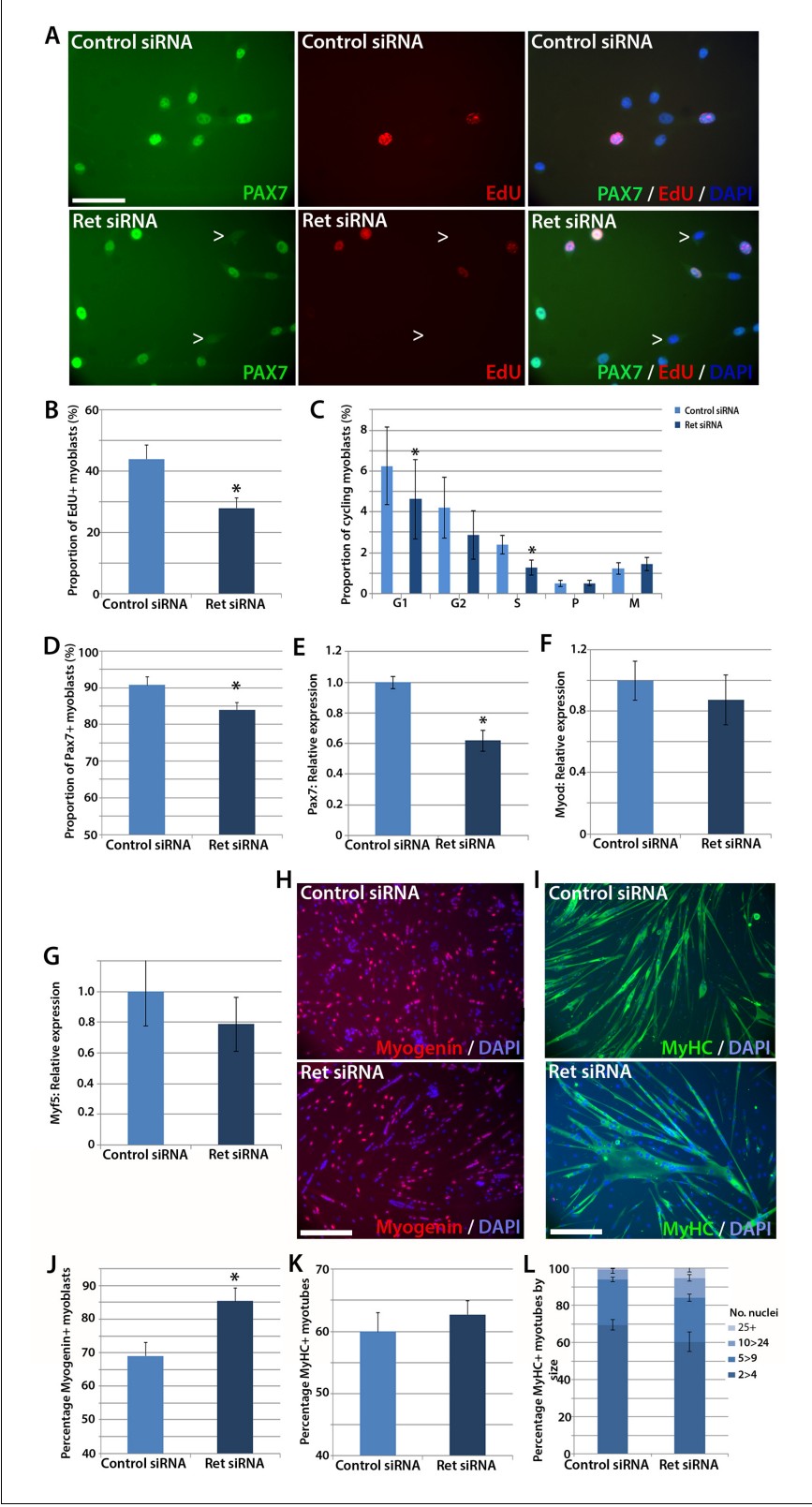

**Figure 3.** Knockdown of *Ret* enhances myogenic differentiation of satellite cells. (A) Satellite cell-derived myoblasts transfected with control or *Ret* siRNA for 48 hr and pulsed for 2 hr with EdU (red) to measure the proliferation rate, and immunolabelled for Pax7 (green) and counterstained with DAPI (blue), quantified in (B and D). (C) Quantification of satellite cell-derived myoblasts transfected with control or *Ret* siRNA and labelled with phospho-Histone H1/H3. Cells were grouped according to their cell cycle stage. (E–G) RT-qPCR analysis of the relative expression of *Pax7*, *MyoD* and *Myf5* in

*Figure 3 continued on next page*

*Figure 3 continued*

satellite cell-derived myoblasts transfected with control or Ret siRNA for 48 hr, with expression normalised to *Gapdh*. (**H** and **I**) Satellite cell-derived myoblasts transfected with *Ret* or control siRNA and incubated in differentiation medium for 48 hr, immunolabelled with Myogenin (red) and counterstained with DAPI (blue) (**H**) or Myosin Heavy Chain (MyHC - green) and counterstained with DAPI (blue) (**I**), quantified in (**J**–**L**). (**L**) Quantification of the relative proportion of small (2 > 4 nuclei), medium (5 > 9 nuclei), large (10 > 24 nuclei) and very large (25+ nuclei) myotubes. Data is mean ± SEM from 3–4 mice in each case, where statistical difference (p<0.05) from control siRNA was determined using a paired Student's *t*-test and denoted by an asterisk. Scale bars equal 50 μm (**A**) and 200 μm (**H** and **I**).

## Knockdown of Gfrα2 or Gfrα4 recapitulates aspects of Ret knockdown

To determine which Gfrα co-receptors expressed in satellite cell-derived myoblasts may activate Ret signalling, myoblasts were transfected with siRNA against Gfrα1, Gfrα2, Gfrα4, or scrambled control siRNA, for 48 hr, before cells were immunolabelled for phospho-Histone H1/H3, Pax7, Myogenin and MyHC (*Figure 4A–D*). Phospho-Histone H1 and H3 immunostaining revealed that knockdown of Gfrα1, Gfrα2 and Gfrα4 were associated with fewer cells in the active stages of the cell cycle (*Figure 4A*). As with *Ret* siRNA, knockdown of Gfrα4 (but not Gfrα1 or Gfrα2) was associated with

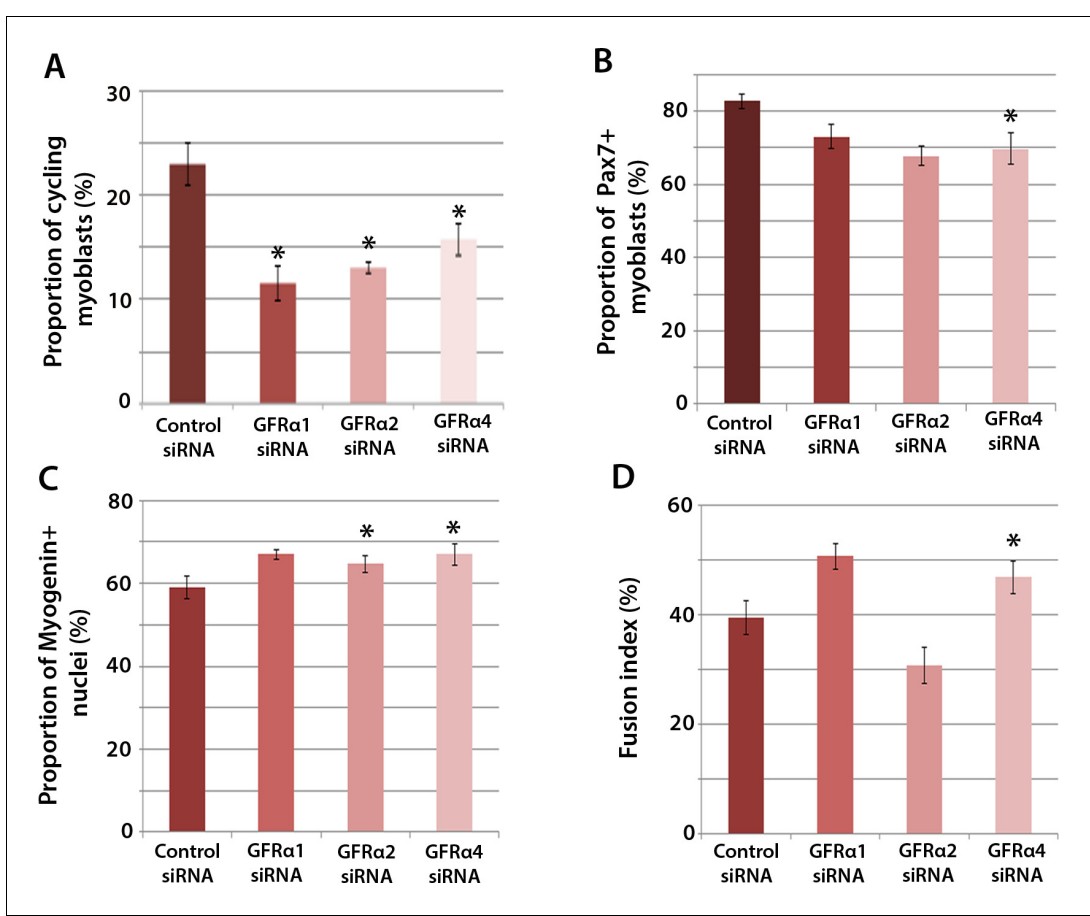

**Figure 4.** Gfrα1, Gfrα2 and Gfrα4 affect myoblast proliferation and myogenic differentiation. (**A**–**D**) Quantification of satellite cell-derived myoblasts transfected with control or Gfrα1, Gfrα2 or Gfrα4 siRNA for 48 hr in proliferation medium (**A** and **B**) or differentiation medium (**C** and **D**). Immunolabelling of cells was performed to quantify the proportion in the cell cycle using anti-phospho-Histone H1 and H3 (**A**), the proportion containing immunosignal for Pax7 (**B**), Myogenin (**C**) and the proportion of nuclei in multinucleated myotubes (fusion index) (**D**). Data is mean ± SEM from 3 independent experiments using 3 mice, where statistical difference (p<0.05) from control siRNA was assessed using a paired Student's *t*-test and denoted by an asterisk.

fewer Pax7-containing satellite cell-derived myoblasts in proliferation medium, from 82.8 ± 2.0% in controls to 69.8 ± 4.3% in cells with siRNA to Gfrα4 (*Figure 4B*). After 48 hr in differentiation medium, the proportion of Myogenin-positive nuclei was significantly increased following knock-down of Gfrα2 and Gfrα4 (*Figure 4C*). Additionally, knockdown of Gfrα4 significantly increased the fusion index (from 39.3 ± 3.1% in the control sample to 46.8 ± 3.0% with Gfrα4 siRNA (*Figure 4D*). Taken together, knockdown of Gfrα co-receptors recapitulated aspects of *Ret* knockdown, indicating that multiple co-receptors may signal through Ret in muscle stem cells.

## Active Ret signalling drives myoblast proliferation but does not affect differentiation

We next used retroviral-mediated constitutive expression to examine the effects of increased *RET* expression on satellite cell proliferation and differentiation. Human *RET9* and *RET51*, together with constitutively active (CA) *RET9* and CA *RET51* (containing a Cys634Lys mutation causing ligand-independent dimerisation of the RET receptor [*Mulligan et al., 1993*]), were cloned into a retroviral backbone containing an *IRES-eGFP* to identify transduced cells. These retroviral constructs encoded RET protein of 170kDa as expected (*Figure 5A*) and human *RET* mRNA was present 48 hr after transduction of mouse myoblasts (*Figure 5B*). Expression of endogenous murine *Ret* was unaltered by human *RET* expression (*Figure 5C*).

To ascertain whether RET affects proliferation rate, myoblasts were infected with retroviruses encoding the RET isoforms and pulsed with EdU for 2 hr in proliferation medium. Constitutive expression of either wild-type RET9 or RET51 did not alter the proportion of eGFP+/EdU+ myoblasts compared to transduction with control retrovirus. However, the CA RET constructs significantly increased proliferation rate, with EdU incorporation rising from 47.6 ± 0.8% in controls to 56.9 ± 1.8% (CA RET9) and 59.2 ± 1.7% (CA RET51) (*Figure 5D and E*). RET did not alter the overall proportion of Pax7-containing cells (*Figure 5F and G*).

To determine whether constitutive expression of RET affects myogenic differentiation, we analysed expression of Myogenin and MyHC in myoblasts cultured in differentiation medium (*Figure 5H–K*). After 24 hr of differentiation, CA RET51 reduced the proportion of myoblasts that contained Myogenin (49.5 ± 1.2%) relative to controls (58.9 ± 3.1%), but by 48 hr there was no longer any difference (*Figure 5H and I*). At this stage, there was no alteration in fusion index (*Figure 5J and K*).

## DUX4 induces Ret in satellite cell-derived myoblasts

To identify genes affected by DUX4 expression, we analysed our microarray data (GSE77100, http://www.ncbi.nlm.nih.gov/geo/query/acc.cgi?acc=GSE77100), which measured transcriptional changes in murine satellite cell-derived myoblasts 20 hr after transduction with retroviruses encoding DUX4, DUX4c (a truncated homologue of DUX4 containing the same homeodomains, encoded by a D4Z4 unit centromeric to the macrosatellite [*Bosnakovski et al., 2008b*]), truncated (tMALDUX4), constitutively active (tMALDUX4-VP16) or dominant-negative (tMALDUX4-ERD) DUX4 constructs (*Banerji et al., 2015*). *Ret* expression was increased 3.45-fold in DUX4-expressing satellite cell-derived myoblasts, and approximately 2 fold by either tMALDUX4-VP16 or tMALDUX4-ERD, relative to controls. Expression of Ret co-receptors *Gfrα2, Gfrα3* and *Gfrα4* were unaltered, although *Gfrα1* expression was reduced 0.57-fold by DUX4 (*Figure 6A*). Expression of DUX4c did not alter *Ret* or *Gfrα* co-receptor expression (*Figure 6A*).

To confirm that *Ret* is a downstream DUX4 target gene in satellite cell-derived myoblasts, we transduced myoblasts with retroviral constructs encoding DUX4 or DUX4c. There was a trend for DUX4 to increase *Ret* transcription after 24 hr (p=0.052), which increased significantly (97.5-fold) after 48 hr, relative to control plasmid-transduced cells (*Figure 6B*). Using isoform-specific primers, we found that DUX4 increased expression of *Ret9* (150-fold) more than *Ret51* (15.8-fold) after 48 hr (*Figure 6C and D*). Using inducible iC2C12-DUX4 and iC2C12-DUX4c myoblasts (*Dandapat et al., 2013*), expression of *Ret* was significantly increased after 12 hr of induction with 200ng/ml doxycycline in the iC2C12-DUX4 line, compared to un-induced myoblasts, and maintained at elevated levels at 24 and 48 hr of induction (*Figure 6E*). Induction of DUX4c in the iC2C12-DUX4c line did not generally alter *Ret* expression (*Figure 6F*). DUX4 activation of *Ret* also led to the detection of

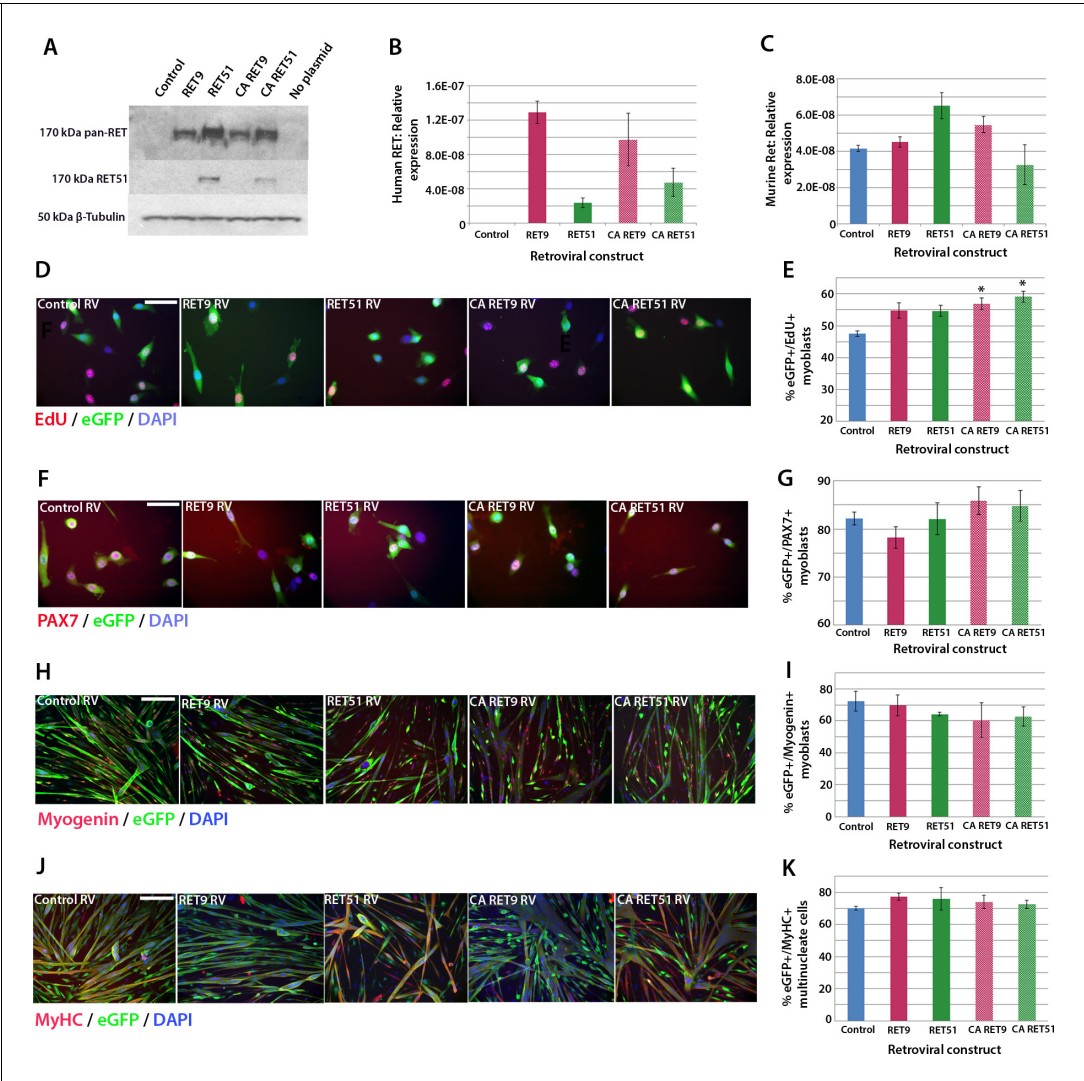

**Figure 5.** Active RET signalling drives satellite cell-derived myoblast proliferation. (**A**) Immunoblot analysis of the RET protein in HEK 293T cells transfected with plasmids encoding human *RET 9, RET 51, CA RET9* and *CA RET51* for 24 hr. Equal quantities of total protein extracts were run on 4–20% gradient PAGE-SDS gels, transferred and blotted with antibodies recognising both RET isoforms (pan-RET, top band and RET51, middle band). Bands of 170 kDa were detected in HEK 293T cells transfected with *RET*-encoding plasmids, but not in cells transfected with control plasmid. β-Tubulin was used as a loading control. (**B–K**) Satellite cell-derived myoblasts transduced with control retrovirus encoding *eGFP* alone, or retroviruses encoding *RET 9, RET 51, CA RET9* or *CA RET51*, together with GFP. mRNA was prepared (**B–C**) or myoblasts immunolabelled for eGFP (to identify transduced cells) and assayed for EdU incorporation (**D** and **E**) or co-immunolabelled for eGFP and either Pax7 (**F** and **G**), Myogenin (**H** and **I**) or MyHC (**J–K**). (**B–C**) RT-qPCR to measure expression of the human versions of RET (**B**) and endogenous murine *Ret* (**C**). (**D** and **E**) Quantification of the proportion of eGFP+ satellite cell-derived myoblasts incorporating EdU after a 2 hr exposure. (**F–K**) Quantification of the proportion of eGFP-expressing satellite cell-derived myoblasts containing Pax7 (**F** and **G**), Myogenin (**H** and **I**) and the fusion index (**J** and **K**). RT-qPCR and quantification of immunolabelling is presented as mean ± SEM from 3 or 4 independent experiments using 3–4 mice, where statistical difference (p<0.05) to transduction with control retrovirus was assessed using a paired Student's *t*-test and denoted by an asterisk. Scale bar equals 50 μm (**D** and **F**) or 200 μm (**H** and **J**).

membrane-located Ret protein, as revealed by immunolabelling with an antibody to Ret51 in DUX4-infected myoblasts (*Figure 6G*).

## Down-regulation of Ret rescues DUX4-mediated inhibition of myogenic differentiation

Our findings that *Ret* is a DUX4 target that influences satellite cell proliferation and differentiation, make it a good candidate to test inhibitors for ameliorating DUX4-induced pathogenesis. A major

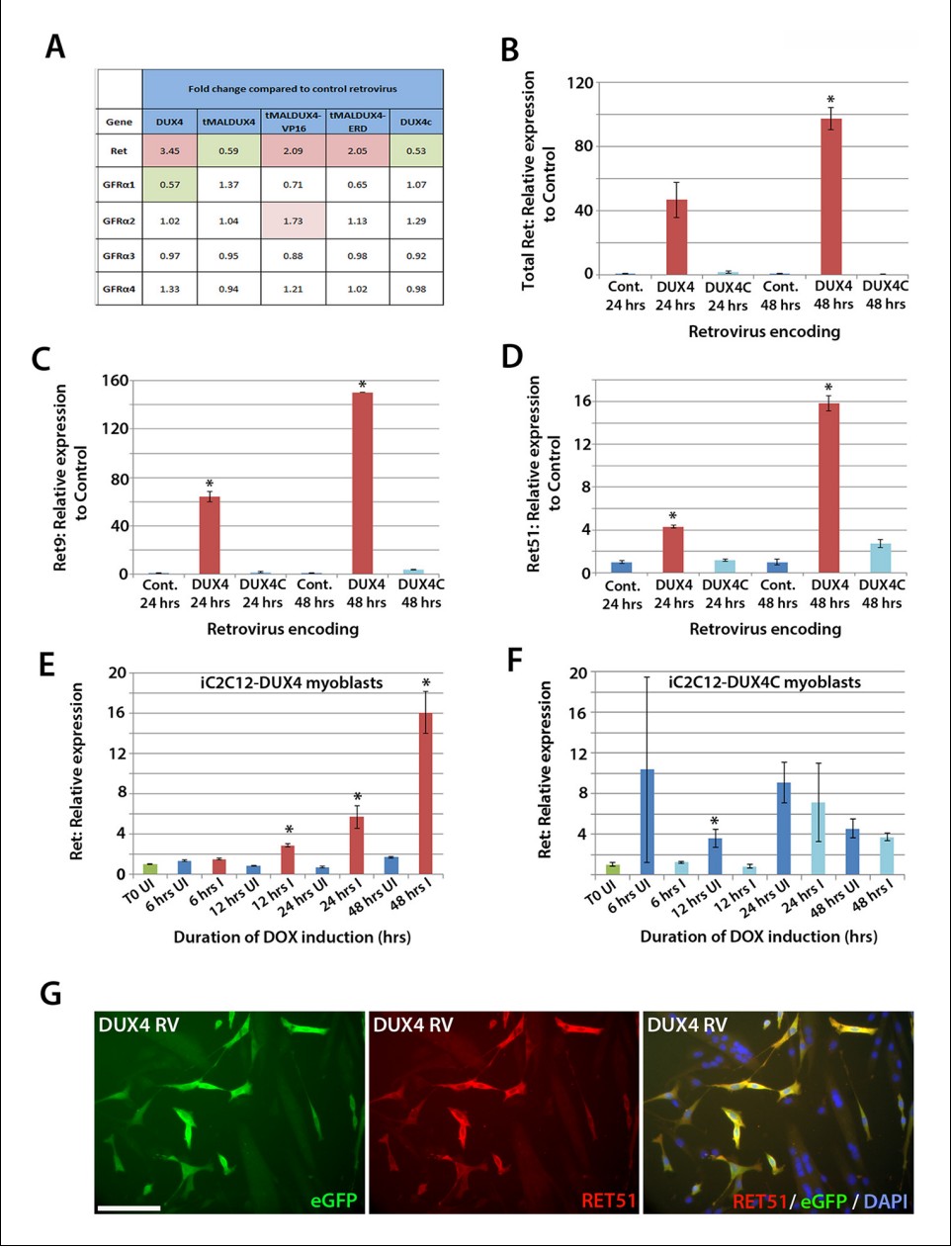

**Figure 6.** DUX4 expression induces *Ret* in satellite cell-derived myoblasts. (**A**) Microarray data of expression levels of Ret and Ret co-receptors Gfrα1–4 in murine satellite cell-derived myoblasts transduced for 20 hr with retroviruses encoding either DUX4, truncated DUX4 (tMALDUX4), constitutively active DUX4 (tMALDUX4-VP16), dominant-negative DUX4 (tMALDUX4-ERD) or DUX4c (*Banerji et al., 2015*). Red highlights increased expression while green highlights reduced expression (fold change) compared to transduction with control retrovirus. (**B–D**) Quantification of total *Ret, Ret9* and *Ret51* expression by RT-qPCR in satellite cell-derived myoblasts transduced with DUX4, DUX4c or control retroviruses at 24 and 48 hr post-infection. (**E–F**) *Ret* expression in iC2C12-DUX4 myoblasts (**E**) and iC2C12-DUX4c myoblasts (**F**) following 200ng/ml doxycycline (DOX) induction. (**G**) Satellite cell-derived myoblasts transduced with DUX4-encoding retrovirus for 24 hr and immunolabelled for eGFP (green) to identify transduced cells and anti-RET51 (red), with a DAPI counterstain (blue). Data is mean ± SEM from 3 independent experiments using 3 mice for (**B–D**) where statistical difference (p<0.05) from transduction with control retrovirus was assessed using a paired Student's *t*-test and denoted by an asterisk. For **E** and **F** , unpaired Student's t-tests were used to assess significance (p<0.05) compared to uninduced cells at each time point. UI = un-induced, I = induced by doxycycline. Scale bar equals 100 µm (**G**).

feature of DUX4 expression in myoblasts is the induction of p53-dependent apoptosis (*Bosnakovski et al., 2008a*; *Kowaljow et al., 2007*; *Wallace et al., 2011*; *Wuebbles et al., 2010*). Interestingly, Ret is a dependence receptor, triggering apoptosis in the absence of its ligands (*Graf et al., 2007*). Therefore, one might hypothesise that DUX4-mediated up-regulation of *Ret* in the absence of sufficient GFLs could lead to cell death.

RT-qPCR revealed that both *Ret* isoforms were expressed in primary murine satellite cells (*Figure 7A*). To test whether over-expression of *RET* isoforms can promote cell death, satellite cell-derived myoblasts were transfected, rather than transduced, with RET plasmids, to induce high levels of expression. Then the relative levels of active caspase 3/7 were measured in proliferating myoblasts using a luminescent assay. However, over-expression of either RET isoform significantly reduced caspase 3/7 release, implying that DUX4-mediated up-regulation of RET does not contribute to the apoptotic phenotype (*Figure 7B*).

DUX4 also suppresses *Pax7* and myogenic regulatory factor (MRF) gene expression and inhibits proliferation and myogenic differentiation (*Mitsuhashi et al., 2013*; *Bosnakovski et al., 2008a*; *Knopp, 2011*; *Xu et al., 2014*). To determine if we could rescue DUX4-induced pathology by inhibiting Ret signalling, we measured the effects of siRNA-mediated inhibition of *Ret* in DUX4-expressing murine satellite cell-derived myoblasts. Myoblasts transduced with control or DUX4 encoding retrovirus were additionally transfected with 20 nM control or *Ret* siRNA for 48 hr in proliferation medium and pulsed with EdU for two hours (*Figure 7C*). As expected, the proportion of transduced eGFP-expressing myoblasts incorporating EdU was significantly reduced when either DUX4 was expressed or when *Ret* was knocked down via siRNA transfection. Transfection of *Ret* siRNA into DUX4-expressing cells did not significantly alter the proportion of cells incorporating EdU relative to control siRNA treated cells. Knockdown of *Ret* did not rescue the reduced number of eGFP+/Pax7+ myoblasts in the presence of DUX4 (*Figure 7D*). Together, this suggests that *Ret* knockdown is unable to rescue the DUX4-mediated proliferation defect in murine primary myoblasts.

We then tested whether knockdown of *Ret* by siRNA could rescue myoblast differentiation in the presence of DUX4 (*Figure 7E*). To assess significant rescue, to compensate for any biological variability and to identify interaction effects, we fitted the data to a binomial model. There was a slight increase in the fusion index of myoblasts expressing control retrovirus and transfected with siRNA against *Ret* relative to control siRNA (p=3.06 $\times$ 10$^{-4}$, *Figure 7E, F and G*, *Figure 7—source data 1*). Myoblasts expressing DUX4 (GFP+) had a dramatically reduced fusion index when transfected with control siRNA (p=2.71 $\times$ 10$^{-65}$). In contrast, *Ret* knockdown rescued myoblast fusion in the presence of DUX4, relative to cells expressing DUX4 and transfected with a control siRNA (p=2.15 $\times$ 10$^{-21}$). Therefore, *Ret* knockdown significantly improves myogenic differentiation in the presence of DUX4.

## Sunitinib, a clinically-approved kinase inhibitor, suppresses Ret signalling to rescue myogenic differentiation

Next, we evaluated small molecule inhibitors of RET for their ability to affect myogenesis in the presence of DUX4. Three RTK inhibitors that block RET phosphorylation were tested: Zactima (Vandetanib/ZD6474) (*Carlomagno et al., 2002*), TG101209 (*Pardanani et al., 2007*) and Sunitinib (Sutent, SU11248) (*Kim et al., 2006*). Zactima and Sunitinib are both clinically approved as therapeutic agents for treating cancers arising from over-activation of RET signalling (*Durante et al., 2013*; *Widmer et al., 2014*).

Since Zactima, TG101209 and Sunitinib also inhibit several RTKs other than RET, including *VEGFR*, *EGF* and *MET* (*Plaza-Menacho et al., 2014*), we first determined the dose needed to inhibit RET signalling and rescue differentiation in murine myoblasts. We tested different concentrations of each drug on CA RET51-expressing murine C2C12 myoblasts, since we found that CA RET51 inhibits myotube formation in this immortalised myoblast line. Thus, fusion in the presence of CA RET51 would indicate that the drug inhibits this constitutively-active human RET mutant. Statistical models revealed that all inhibitors significantly rescued muscle differentiation in CA RET51-expressing C2C12s (*Figure 8A–C*, Sunitinib: p=2 $\times$ 10$^{-16}$, TG101209: p=2 $\times$ 10$^{-16}$, Zactima: p=4.18 $\times$ 10$^{-14}$, *Figure 8—source data 1*). This occurred in a dose-dependent manner with Sunitinib, and to an extent, with TG101209. Crucially, this dose-dependent response to Sunitinib and TG101209 occurred without overtly affecting cell number (*Figure 8D–F*, *Figure 8—source data 2*). Control C2C12 myoblasts were unaffected by Sunitinib and TG101209 at low doses (up to 250 ng/ml), but at higher doses, showed a small increase in fusion index. In contrast, Zactima significantly affected cell

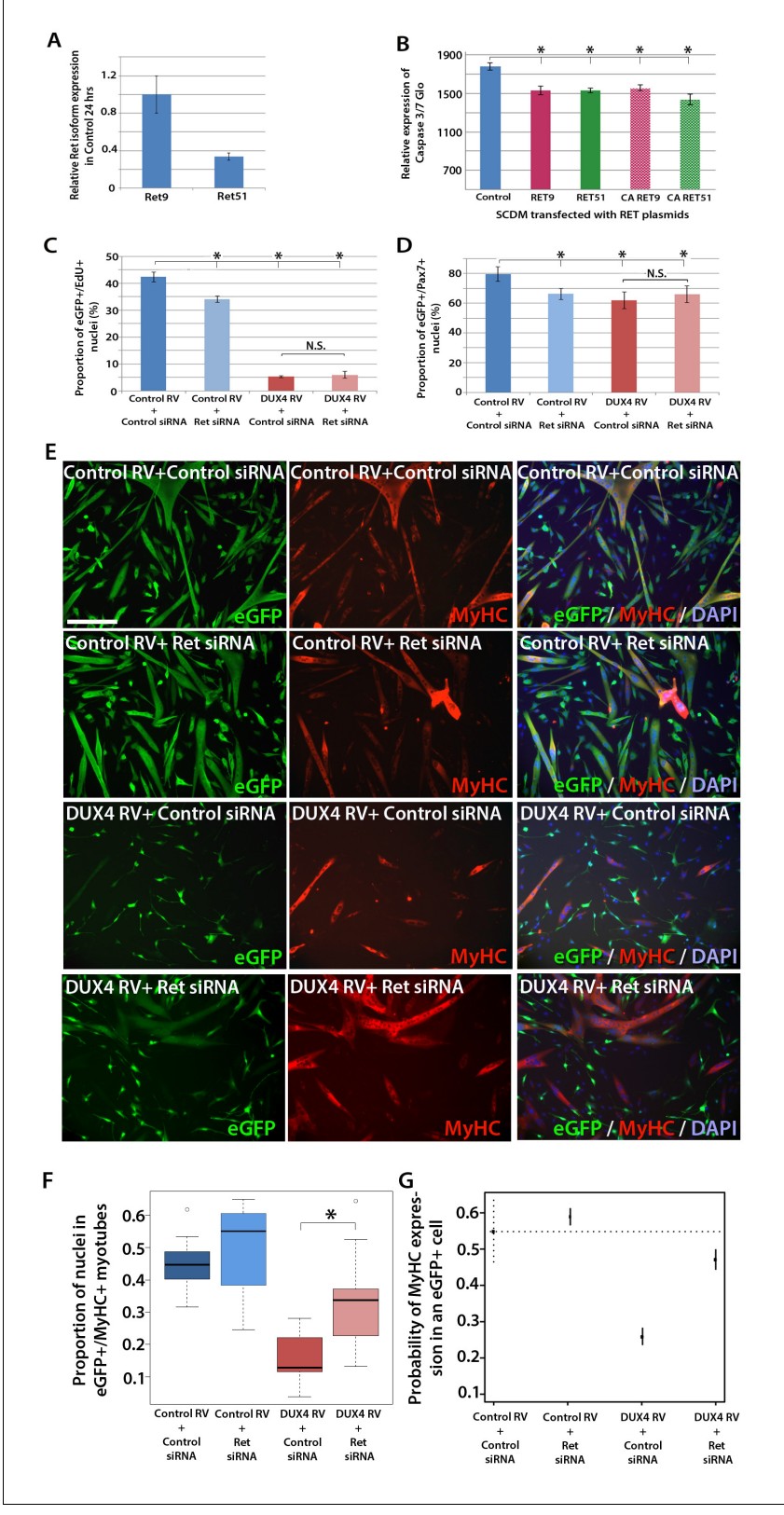

**Figure 7.** Knockdown of *Ret* rescues DUX4-mediated inhibition of myogenic differentiation. (**A**) Relative expression of *Ret9* and *Ret51* in myoblasts infected with control retrovirus after 24 hr. (**B**) Quantification of a caspase 3/7 Glo assay to measure apoptosis in satellite cell-derived myoblasts transfected with control plasmid encoding *GFP*, or *RET 9*, *RET 51*, *CA RET9* or *CA RET51* encoding plasmids. (**C–D**) Quantification of proliferation by EdU

*Figure 7 continued on next page*

*Figure 7 continued*

incorporation (C) and *Pax7* expression (D) in *eGFP*-expressing myoblasts transduced with control or DUX4-encoding retrovirus and then transfected with control or *Ret* siRNA for 48 hr. Data is mean ± SEM from 3 mice where statistical difference (p<0.05) to control plasmid (B) or control retrovirus + control siRNA (C and D) was assessed using a paired Student's *t*-test and denoted by an asterisk. Bar is comparison between indicated conditions, where N.S. denotes non-significant difference. (E) Immunolabelling of satellite cell-derived myoblasts transduced with control or *DUX4*-encoding retrovirus and transfected with control or *Ret* siRNA following culture for 24 hr in differentiation medium. Transduced cells were detected by immunolabelling for eGFP and examined for MyHC to identify terminally differentiated myotubes. (F) The fusion index was calculated for eGFP-positive cells expressing control or *DUX4*-encoding retrovirus and transfected with control or *Ret* siRNA. Bar represents statistical test significance, where an asterisk denotes p<0.01. (G) Plot of probability that a cell has MyHC immunoreactivity derived from binomial models (*Figure 7—source data 1*). Error bars represent 95% confidence intervals. Four replicates were counted for each condition and repeated using 3 mice. Scale bar equals 200 μm (E).
The following source data is available for figure 7:

**Source data 1.** Maximum likelihood parameters for a logistic model containing an interaction term, and a random effect term describing the probability of a nucleus being present in a MyHC+ cell.

number in a non-dose dependent manner and had a strong effect on fusion in control samples (*Figure 8C and F* and *Figure 8—source data 2*). Sunitinib was selected for further analysis, as it promoted an effective rescue of fusion in the presence of CA RET51, without causing large changes to cell number (*Figure 8G*).

## Sunitinib suppresses DUX4-mediated pERK1/2 signalling

To address the mechanism of Sunitinib in myoblasts, we examined characterised downstream targets of Ret signalling, namely phosphorylated ERK1/2 (pERK1/2) and phosphorylated AKT (pAKT) e.g. (*Mograbi et al., 2001*). DUX4 was induced in iC2C12-DUX4 myoblasts using 500 ng/ml doxycycline with or without the addition of 250 ng/ml Sunitinib for 20 hr (*Figure 8H*). Western blot analysis showed that Sunitinib significantly reduced pERK1/2 levels in DUX4-expressing myoblasts, without affecting total ERK (*Figure 8H and I*). Total AKT and pAKT levels were unaltered by Sunitinib for 20 hr in iC2C12-DUX4 myoblasts expressing DUX4 (*Figure 8H and J*).

## Sunitinib does not rescue proliferation or apoptosis in DUX4-expressing murine myoblasts

We first assessed how 250 ng/ml Sunitinib affected proliferation in DUX4-expressing primary murine satellite cells by measuring EdU incorporation (*Figure 9A*). There was no significant difference in proliferation of control cells treated with DMSO vehicle control (43.5 ± 2.0%) or Sunitinib (43.3 ± 0.7%). Proliferation in DUX4-expressing myoblasts was also unaffected by Sunitinib (8.6 ± 2.0%) compared to DUX4-expressing cells treated with DMSO (8.7 ± 1.7%, p<0.05) (*Figure 9A* and *Figure 9—source data 1*).

DUX4 expression in myoblasts also leads to repression of MyoD (*Bosnakovski et al., 2008a*; *Snider et al., 2009*). Treatment with Sunitinib rescued the number of MyoD-containing myoblasts in the presence of DUX4 when analysed by a statistical model (p=3.8 × 10$^{-4}$), although the relative change was not large (*Figure 9B* and *Figure 9—source data 2*).

Additionally although expression of Ret in satellite cell-derived myoblasts was not apoptotic (*Figure 7B*), it was important to understand whether Sunitinib affects apoptosis in DUX4-expressing cells. iC2C12-DUX4 myoblasts were induced with 250 ng/ml and 500 ng/ml doxycycline for 24 hr, which resulted in apoptosis (*Figure 9C*), and exposed to either 250 ng/ml or 500 ng/ml Sunitinib. Live cells were stained with fluorescent-conjugated apoptotic markers Annexin V (AV) and propidium iodide (PI), and analysed by flow cytometry (*Figure 9C*). While DUX4 expression increased the proportion of iC2C12-DUX4 myoblasts gated as apoptotic (AV+/PI−, AV−/PI+ and AV+/PI+), treatment with Sunitinib had no significant effect (*Figure 9D*).

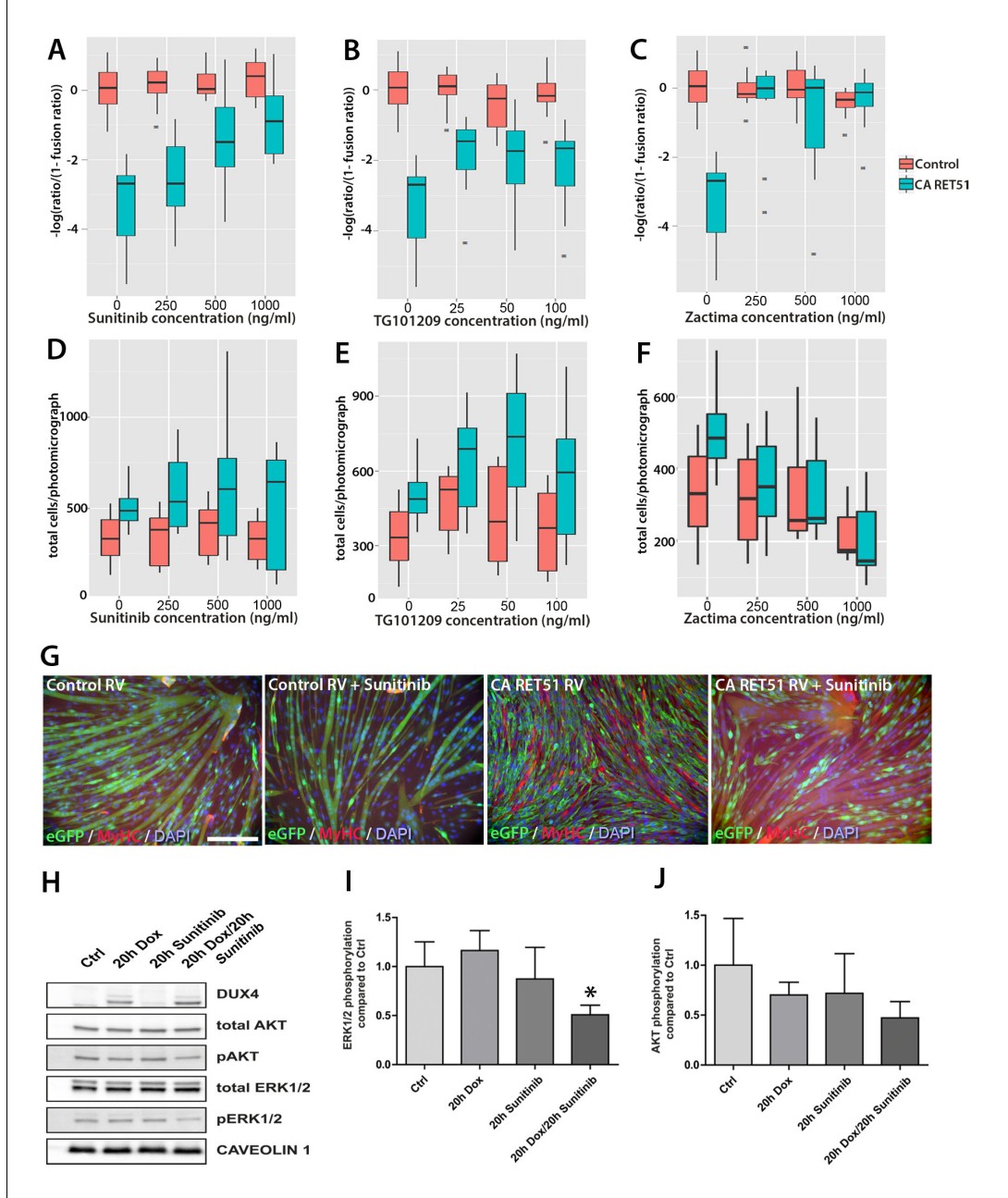

**Figure 8.** Sunitinib blocks the RET-induced phenotype in murine myoblasts. (A–C) Quantification of the fusion index of C2C12 myoblasts transduced with control (red) or *CA RET51*-encoding retrovirus (blue) and treated with Sunitinib, TG101209 or Zactima at varying doses is shown as a statistical model fitted to the –log of the odds ratio of the fusion index (-log ratio/(1-ratio)) (*Figure 8—source data 1*). (D–F) Quantification of the number of C2C12 myoblasts transduced with control (red) or *CA RET51*-encoding retrovirus (blue) when treated with Sunitinib, TG101209 or Zactima at varying doses (*Figure 8—source data 2*). (G) C2C12 myoblasts transduced with control or *CA RET51*-encoding retrovirus and induced to differentiate for 60 hr in the presence of either 1 µg/ml Sunitinib or DMSO vehicle control, before co-immunolabelling with antibodies to eGFP (green) to detect transduced cells and MyHC (red), with a DAPI nuclear counterstain (blue). All quantification represents an average of three independent experiments. Scale bar equals 200 µm. (H) Representative immunoblot of proteins extracted from un-induced iC2C12-DUX4 myoblasts, or cells induced with 500ng/ ml doxycycline (Dox) to express DUX4, 250 ng/ml Sunitinib or both doxycycline and Sunitinib. Probed with antibodies against DUX4, total AKT, phosphorylated (p) AKT, total ERK1/2 and phosphorylated (p) ERK1/2, with Caveolin-1 used as a loading control. All bands shown were visualised on the same membrane. (I and J) Protein band intensity was quantified with the ChemiDoc MP System and normalised to housekeeping protein Caveolin-1. The ratios of pERK1/2: total ERK1/2 and pAKT: total AKT in the treated groups were compared to the ratios in the control group. Quantification of the ratio between pERK1/2: total ERK1/2 compared to control shows that Sunitinib suppresses DUX4-mediated pERK1/2 signalling. Data is mean ± SEM from 3 independent experiments, where an asterisk denotes a significant difference (p<0.05) from Control using an unpaired Student's *t*-Test.

*Figure 8 continued on next page*

*Figure 8 continued*

The following source data is available for figure 8:

**Source data 1.** Maximum likelihood parameters for a logistic model containing an interaction term and a random effect (mouse) to describe the effect of CA RET51 (RET51-MENA) expression and Sunitinib, TG101209 or Zactima on fusion in C2C12 myoblasts.

**Source data 2.** Quasi-Poisson model parameters for a fixed-effects factorial model incorporating a parameter to account for the replicate effects (Batch) on the number of cells expressing CA RET51 (RET51-MEN2A) or control when treated with different concentrations of Sunitinib, TG101209 or Zactima.

## Sunitinib rescues the myogenic capacity of DUX4-expressing murine myoblasts

We next determined whether Sunitinib could rescue fusion in murine satellite cell-derived myoblasts expressing DUX4. Myoblasts were cultured at high density in the presence of 125–500 ng/ml of Sunitinib and induced to differentiate for 24 hr prior to immunolabelling with MyHC (*Figure 10A and B*). Fusion of control myoblasts was unaffected by Sunitinib at all doses (quantified in *Figure 10C*). Statistical models showed that the treatment of DUX4-expressing (GFP+) myoblasts with Sunitinib increased fusion index at all doses, relative to cells treated with DMSO vehicle control (p<2 $\times$ 10$^{-16}$, *Figure 10C*, *Figure 10—source data 1*). This also revealed a saturation of the effect of Sunitinib at concentrations above 250 ng/ml. Sunitinib treatment also increased the proportion of large myotubes that formed in the presence of DUX4, and myotubes containing 25+ nuclei were only observed in Sunitinib-treated samples.

To directly determine whether there is an interaction between Sunitinib and DUX4 in murine primary myoblasts that positively affects the fusion index, we next transduced myoblasts with control or DUX4 retrovirus in the presence of DMSO or 250 ng/ml Sunitinib and induced them to differentiate for 24 hr at high density (*Figure 10D*). We then used a logistic model to evaluate any interaction (*Figure 10E*, *Figure 10—source data 2*). This revealed that the effect of Sunitinib on fusion is dependent on the presence of DUX4 (p=2 $\times$ 10$^{-16}$) and that there is a significant recovery of fusion in the presence of DUX4 that is due to addition of Sunitinib (p=2.3 $\times$ 10$^{-7}$).

## Sunitinib enables DUX4-expressing murine myoblasts to differentiate independently of fusion

Using a retroviral expression system to express DUX4 resulted in a transduction rate of approximately 40–50%. However, this approach does not allow determination of whether the increased rate of fusion of DUX4-expressing cells in the presence of Sunitinib was due to Sunitinib rescuing the differentiation defect directly, or by simply enabling DUX4-expressing myoblasts to fuse to control, untransduced myoblasts. To determine whether DUX4-expressing (GFP+) myoblasts could differentiate independently of fusion, we seeded myoblasts at low density before treating with 250 ng/ml Sunitinib and inducing differentiation for 24 or 48 hr. Samples were then analysed on the basis of individual differentiated myocytes using immunolabelling to detect Myogenin and MyHC protein (*Figure 10F and G*).

Sunitinib treatment on control-transduced samples did not change the proportion of myoblasts expressing Myogenin (27.4 ± 1.1%), relative to controls (27.6 ± 0.9%) (p=0.86). In contrast, a significant increase (p=0.028) was noted in the proportion of DUX4 expressing (GFP+) myocytes that were Myogenin+ in the presence of Sunitinib (20.8 ± 1.9%) relative to untreated DUX4-expressing myocytes (11.0 ± 0.6%). The proportion of GFP+ myoblasts expressing MyHC was significantly reduced in the presence of DUX4 (2.4 ± 0.6%) relative to control cells (19.4 ± 2.5%, p=0.028) or to cells exposed only to Sunitinib (21.1 ± 2.1%). In the presence of Sunitinib however, the proportion of MyHC+ cells in DUX4-transduced cultures was significantly increased after 24 hr exposure (7.8 ± 1.3%) (p=0.016).

We then used statistical models to evaluate the importance of the interaction between Sunitinib and DUX4 on Myogenin and MyHC expression in myocytes plated at low density (*Figure 10F and G*). Sunitinib had no effect on the number of cells containing Myogenin after 24 or 48 hr in control samples, in contrast to the marked reduction with DUX4 (p<2 $\times$ 10$^{-16}$). The addition of Sunitinib significantly rescued Myogenin expression in DUX4-expressing (GFP+) myoblasts at 24 (p<1.03 $\times$ 10$^{-5}$)

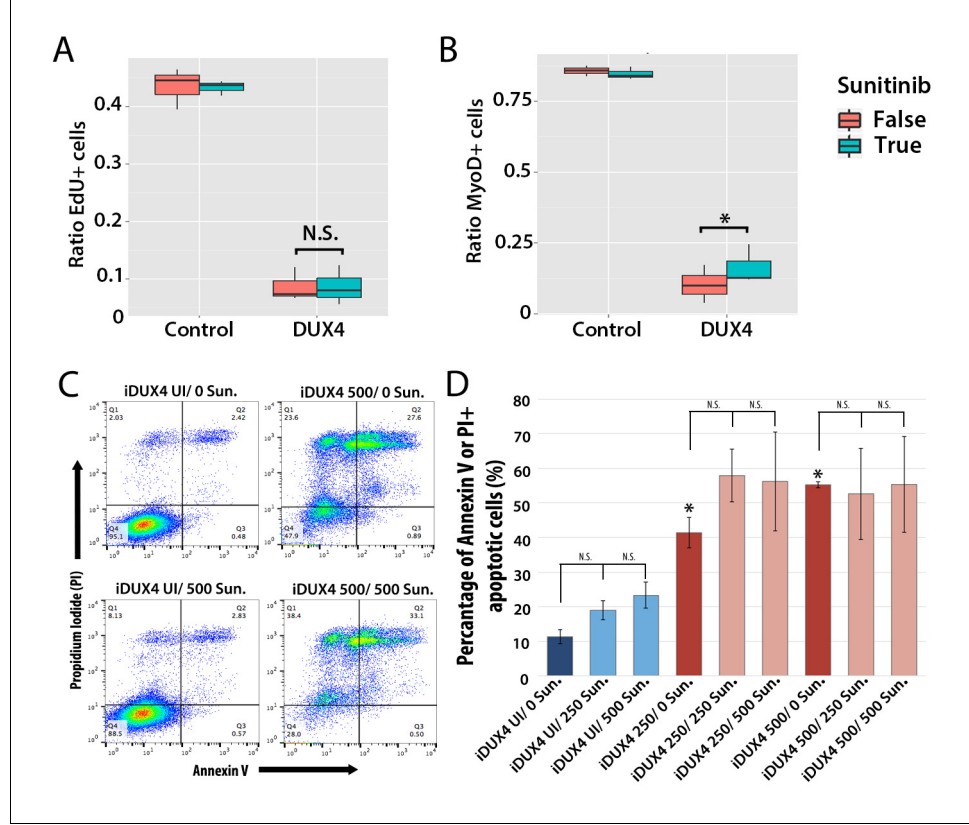

**Figure 9.** Sunitinib does not prevent apoptosis in DUX4-expressing myoblasts. (A–B) Quantification of murine satellite cell-derived myoblasts transduced with control or DUX4-encoding retrovirus and treated with either 250 ng/ml Sunitinib (blue) or DMSO vehicle control (red). Cells were cultured at low density and incubated in proliferation medium prior to immunolabelling to detect eGFP and either EdU (A) or MyoD (B). The ratio represents the proportion of cells with EdU (A and *Figure 9—source data 1*) or MyoD (B and *Figure 9—source data 2*) labelling in the presence (True, blue), or absence of Sunitinib (False, red). Bar represents statistical test significance, where an asterisk denotes p<0.01 or N.S. means non-significant.(C) FACS analysis of murine iC2C12-DUX4 myoblasts cultured in control medium (UI) or induced to express DUX4 with 250 ng/ml or 500 ng/ml doxycycline and untreated (0), or exposed to 250 ng/ml or 500 ng/ml Sunitinib for 24 hr. (D) Quantification of the mean proportion of apoptotic iC2C12-DUX4 myoblasts based upon the expression of Annexin V and propidium iodide (PI). As a control, an asterisk denotes increased apoptosis by DOX-mediated induction of DUX4 compared to un-induced cells (all without Sunitinib treatment). An average of 3 independent experiments, where statistical difference to myoblasts not exposed to Sunitinib at each concentration of doxycycline was tested using an unpaired two-tailed Student *t*-test, where p<0.05 represents significance (*).

The following source data is available for figure 9:

**Source data 1.** Maximum likelihood parameters for a logistic model containing an interaction term, and a random effect term (the mouse) that describes the proportion of myoblasts transduced with DUX4 or control retrovirus and incorporating EdU when exposed to Sunitinib or DMSO.

**Source data 2.** Maximum likelihood parameters for a logistic model containing an interaction term, and a random effect term (the mouse) that describes the proportion of cells transduced with DUX4 or control retrovirus and expressing MyoD when exposed to Sunitinib or DMSO.

and 48 hr (p<$1.38 \times 10^{-5}$, *Figure 10F*, *Figure 10—source data 3*). As with Myogenin, Sunitinib significantly increased the proportion of DUX4-expressing (GFP+) cells with MyHC at 24 hr (p=$4.21 \times 10^{-5}$) and 48 hr (p=$7.94 \times 10^{-3}$, *Figure 10G*, *Figure 10—source data 4*), thus Sunitinib improves myogenic differentiation/fusion.

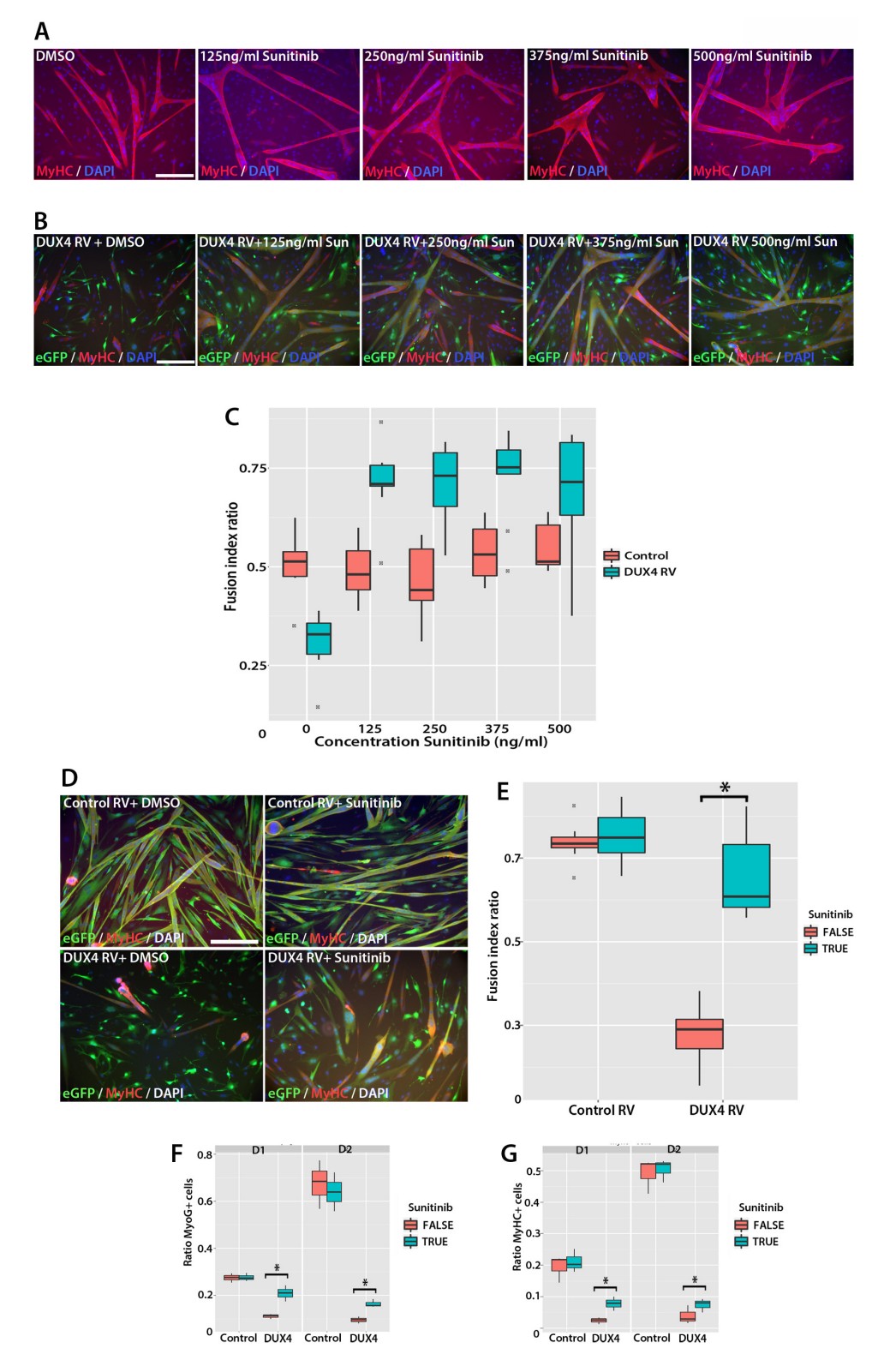

**Figure 10.** Sunitinib rescues differentiation of DUX4-expressing myoblasts. (**A**) Satellite cell-derived myoblasts induced to differentiate in the presence of varying doses of Sunitinib, and immunolabelled for MyHC (red) and counterstained with DAPI (blue). (**B** and **C**) Immunolabelling of satellite cell-derived myoblasts transduced with retrovirus encoding *DUX4* and *GFP* (green) to detect MyHC (red) following treatment with varying doses of Sunitinib in differentiation medium. (**C**) Fusion index calculated as the ratio of nuclei in eGFP+/MyHC+ multinucleate cells/total nuclei (*Figure 10—source data*

*Figure 10 continued on next page*

*Figure 10 continued*

*1*). (**D**) Satellite cell-derived myoblasts transduced with control *GFP* or *DUX4* and *GFP*-encoding retrovirus in the presence of 250 ng/ml Sunitinib or DMSO at high density and induced to differentiate for 24 hr before co-immunolabelling with eGFP (green), MyHC (red) and counterstained with DAPI (blue). (**E**) Fusion index quantified with myoblasts exposed to DMSO (False: red) or 250 ng/ ml Sunitinib (True: blue) (*Figure 10—source data 2*). (**F** and **G**) Quantification of satellite cell-derived myoblasts transduced with control *GFP* or retrovirus encoding *DUX4* and *GFP* and treated with either 250 ng/ ml Sunitinib (blue) or DMSO (red). Cultured at low density in differentiation medium for 24 or 48 hr (D1, D2) to form unfused myocytes and co-immunolabelled with eGFP to identify transduced cells and either Myogenin (**F** and *Figure 10—source data 3*) or MyHC (**G** and *Figure 10—source data 4*). All experiments were independently performed 3 times. The statistical significance of differences described in *Figure 10—source data 1– 4*. Bar represents statistical test significance, where an asterisk denotes p<0.01. Scale bar equals 200 μm.

The following source data is available for figure 10:

**Source data 1.** Maximum likelihood parameters for a logistic model containing an interaction term between DUX4 and Sunitinib that describes the fusion index of satellite-cells grown at high density with or without DUX4 transduction.

**Source data 2.** Maximum likelihood parameters for a logistic model containing an interaction term, and a random effect term (the mouse) that describes the fusion index of myoblasts transduced with DUX4 or control retrovirus and grown at high density when exposed to Sunitinib or DMSO.

**Source data 3.** Maximum likelihood parameters for a logistic model containing an interaction term, and a random effect term (the mouse) to describe MyoG expression in satellite cell-derived myoblasts expressing DUX4 or control retrovirus and exposed to Sunitinib or DMSO, when grown at low density.

**Source data 4.** Maximum likelihood parameters for a logistic model containing an interaction term, and a random effect term (the mouse) to describe MyHC expression in satellite cell-derived myoblasts expressing DUX4 or control retrovirus and exposed to Sunitinib or DMSO, when grown at low density.

## Sunitinib treatment improves proliferation and differentiation of FSHD myoblasts

To determine whether Sunitinib improves myogenic differentiation in human FSHD myoblasts, we obtained clonal cell lines derived from a mosaic FSHD1 individual (*Krom et al., 2012*): a control clone containing a 'healthy' number of D4Z4 repeats (54.6) and a clone containing a pathogenic contraction of *D4Z4* units and expressing DUX4 (54.12), thus apart from the number of DUX4 repeats, both have the same genetic background. Although both clones proliferate and undergo differentiation in culture (*Krom et al., 2012*), we found that the 54.12 pathogenic line proliferated at a slower rate, had an eccentric cell shape and differentiated into smaller myotubes than the healthy 54.6 control myoblasts, as revealed by a lower fusion index (*Figure 11*). This phenotype is similar to that reported for primary myoblasts from FSHD patients (*Barro et al., 2010*).

To determine if Sunitinib was able to rescue pathogenic phenotypes in the human FSHD cell line, we treated both cell lines with varying concentrations of Sunitinib in either proliferation or differentiation medium. There was no change in cell shape (eccentricity) of 54.6 control myoblasts in the presence of Sunitinib at any dose (*Figure 11A and C*). While 54.12 FSHD myoblasts treated with vehicle control or the lower doses of Sunitinib (125 or 250 ng/ml) retained a significantly more elongated cell shape relative to control 54.6 myoblasts (p<0.01), myoblasts treated with higher doses of Sunitinib no longer exhibited an abnormal cell shape compared to 54.6 controls (*Figure 11C*, *Figure 11— source data 1*).

Importantly, the reduced proliferation rate of FSHD 54.12 myoblasts compared to 54.6 (p=2.55 × $10^{-5}$) was rescued upon exposure to 250 ng/ml, 500 ng/ml or 750 ng/ml of Sunitinib, concentrations which did not affect control 54.6 myoblasts (*Figure 11A and D*, *Figure 11—source data 2*).

Fusion of control 54.6 myoblasts was increased by Sunitinib at small, but significant, levels in a non-linear manner. In contrast, Sunitinib had a strong and highly significant effect on the fusion index of 54.12 cells at all concentrations tested (p<2 × $10^{-16}$). The fusion index increased in a dose-dependent manner in the presence of 125 or 250 ng/ml of Sunitinib. 250 ng/ml Sunitinib had the most potent effect on fusion of 54.12 myoblasts, achieving a level that was comparable to that of 54.6 myoblasts (*Figure 11B and E*). At doses of 500 or 750 ng/ml Sunitinib, a fusion of 54.12 myoblasts was not enhanced to similar levels as observed at 250 ng/ml, suggestive of inhibitory effects at these higher doses (*Figure 11B and E*, *Figure 11—source data 3*).

Thus, in myoblasts lacking the FSHD1-specific D4Z4 contraction, Sunitinib does not have a large effect on human myoblast morphology, proliferation or ability to form multinucleated myotubes. In

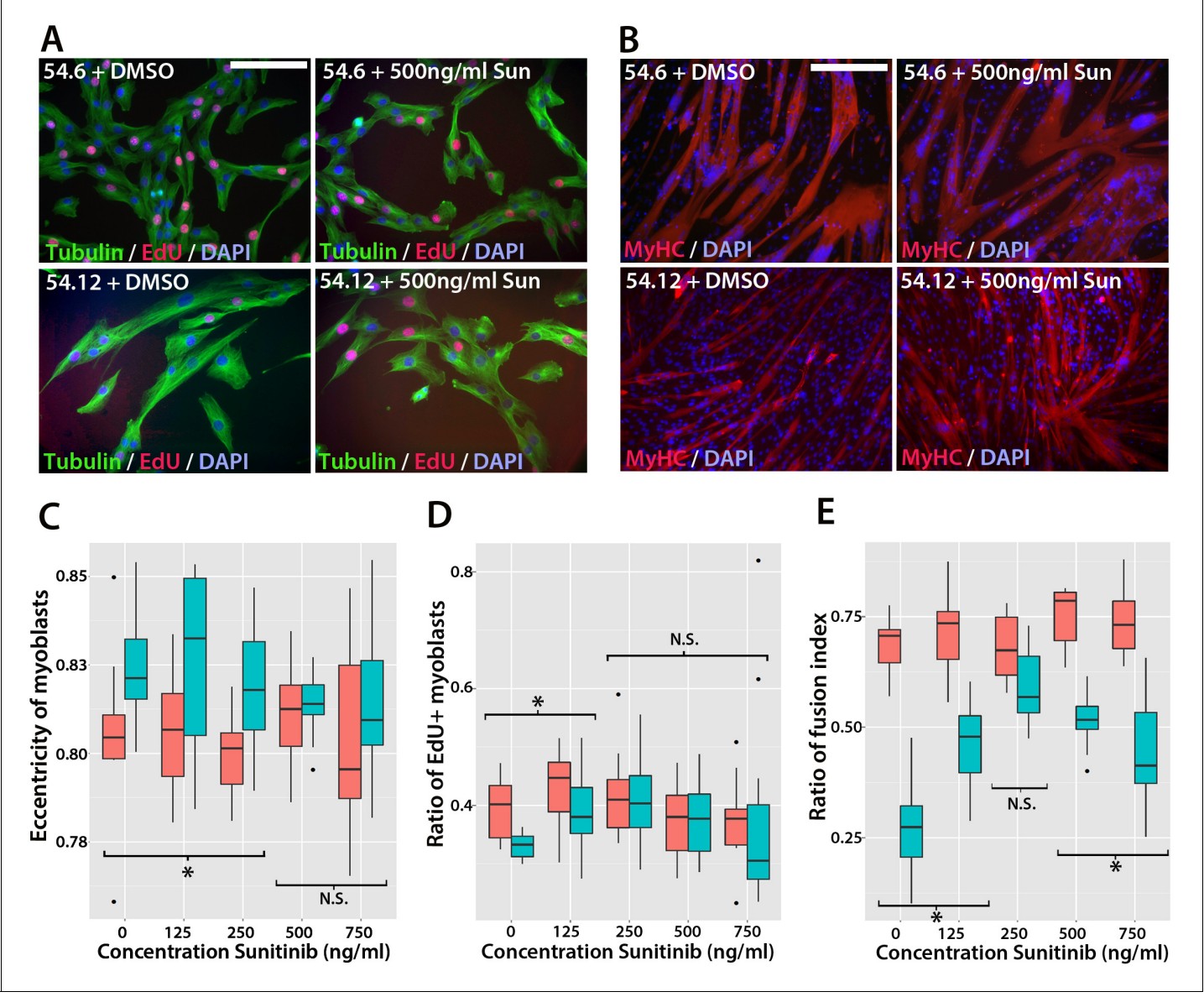

**Figure 11.** Sunitinib improves the pathogenic phenotype of FSHD myoblasts. (A) Immunolabelling of mosaic FSHD patient-derived myoblast cell lines 54.6 (control) and 54.12 (FSHD) exposed to DMSO vehicle control or 500 ng/ml Sunitinib. EdU incorporation (red) was revealed using the Click-iT assay and myoblasts immunolabelled for β-TUBULIN (green) with a DAPI nuclear counterstain (blue). (B) Immunolabelling of MYHC (red) in myoblasts grown at high density in differentiation medium with either DMSO vehicle control or 500 ng/ml Sunitinib. (C–E) Quantification of cell shape (eccentricity) assessed using β-TUBULIN immunolabelling (C and *Figure 11—source data 1*), EdU incorporation (D and *Figure 11—source data 2*) and fusion index (E and *Figure 11—source data 3*) plotted relative to varying Sunitinib concentrations in 54.6 control (red) and 54.12 FSHD (blue) myoblasts. Statistical analysis described in *Figure 11—source data 1–3*. All experiments were independently performed 3 times. Bar represents statistical test significance at each concentration of Sunitinib, where an asterisk denotes p<0.01, while N.S. means non-significant. Scale bars equal 50 µm (A) 200 µm (B).

The following source data is available for figure 11:

**Source data 1.** A linear model that describes the relationship between the shape (eccentricity) of control 54.6 and FSHD 54.12 human myoblasts relative to different doses of Sunitinib.

**Source data 2.** A binomial model that tests the relationship between the proliferation of control 54.6 and FSHD 54.12 human myoblasts with different doses of Sunitinib.

**Source data 3.** A binomial model that tests the relationship between the fusion of control 54.6 and FSHD 54.12 human myoblasts relative to different doses of Sunitinib.

contrast, myoblasts containing the D4Z4 contraction are sensitive to Sunitinib, which alters cell morphology to resemble control myoblasts. Importantly, these FSHD human myoblasts have an increased proliferation rate and greater myogenic differentiation capacity when treated with Sunitinib.

## Sunitinib improves the regenerative capacity of FSHD myoblasts in vivo

We then tested whether Sunitinib could improve the regenerative capacity of human FSHD 54.12 myoblasts in vivo by grafting into regenerating muscles of immunodeficient, NOD/scid/γ-chain

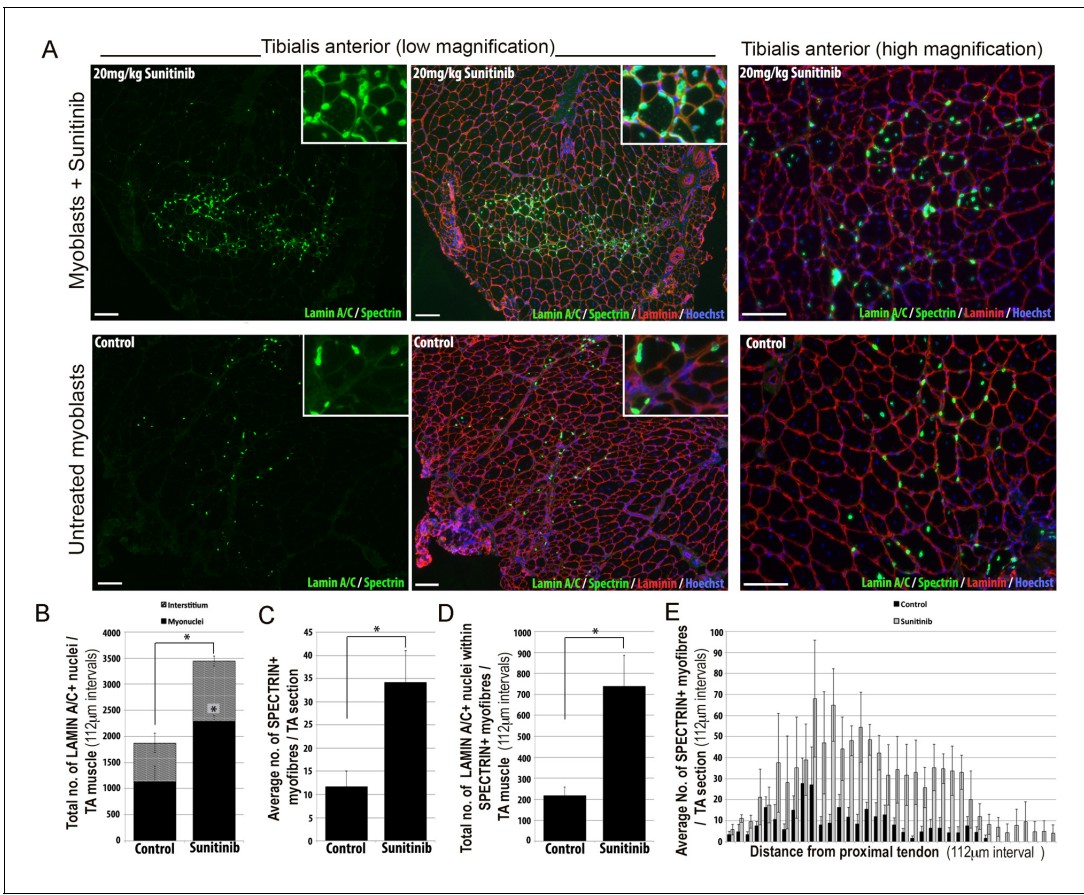

**Figure 12.** Sunitinib improves engraftment and differentiation of human FSHD myoblasts in murine skeletal muscle. (**A**) Representative 8 µm thick cryosections from the tibialis anterior (TA) muscle of immunodeficient mice transplanted with $1 \times 10^6$ human 54.12 FSHD myoblasts per muscle, and dosed with 20 mg/kg/day Sunitinib or PBS (control) for 3 weeks. Immunofluorescence for human-specific LAMIN A/C (green) and SPECTRIN (green) reveal the location and fate of transplanted cells, murine Laminin (red) delimits all myofibres and Hoechst (blue) reveals total nuclei. (**B**) Quantification of the total number of LAMIN A/C cells from each section per TA, subdivided into cells located in the interstitum (grey hatched) and nuclei (myonuclei) within myofibres (black) (also see *Figure 12—source data 1*). (**C**) Quantification of the mean number of human SPECTRIN positive myofibres per TA section (also see *Figure 12—source data 2*). (**D**) Quantification of the total number of human SPECTRIN positive myofibres containing at least one human LAMIN A/C positive cell. (**E**) The mean number of SPECTRIN-positive myofibres per TA section along the length of the muscle. Data is mean ± SEM from 4 control and 4 Sunitinib-treated mice, with statistical significance (p<0.05) tested against control mice using an unpaired Student's t-test and indicated with an asterisk. Further statistical analysis is described in *Figure 12—source data 1* and *2*. Scale bar represents 100 µm.

The following source data is available for figure 12:

**Source data 1.** A binomial model evaluating whether the proportion of LAMIN A/C+ nuclei derived from grafted FSHD 54.12 human myoblasts in a muscle fibre is significantly affected by exposure to Sunitinib.

**Source data 2.** A binomial model evaluating whether the proportion of muscle fibres containing SPECTRIN and LAMIN A/C derived from grafted FSHD 54.12 human myoblasts is significantly affected by exposure to Sunitinib.

knockout mice (*Figure 12*). Mice were pre-treated with an intraperitoneal injection of 20 mg/kg Suni-tinib malate or PBS control for 24 hr prior to cryoinjury of the tibialis anterior (TA) muscle to induce regeneration. TA muscles were then grafted intramuscularly with $1 \times 10^6$ 54.12 FSHD myoblasts, which had been cultured in control medium or pre-treated with 250 ng/ml Sunitinib for 24 hr. Mice were given 20 mg/kg/day Sunitinib or PBS for 3 weeks.

Analysis was performed blinded. After 3 weeks, the TA muscles were removed and serial cryosections cut along the length of the muscle. Cryosections were immunostained with human-specific anti-bodies against LAMIN A/C to quantify donor cell engraftment and SPECTRIN to measure myogenic differentiation of donor myoblasts (*Figure 12A*). There was a significant increase in both the total number of LAMIN A/C positive cells (total from all cryosections along the length of the muscle) (*Figure 12B*, *Figure 12—source data 1*) and the number of LAMIN A/C positive cells located within myofibres (myonuclei), in mice treated with Sunitinib compared to controls (*Figure 12B*). The mean number of SPECTRIN positive myofibres per section (*Figure 12C*, *Figure 12—source data 2*) and the total number of LAMIN A/C positive cells located within SPECTRIN-positive myofibres (*Figure 12D*) were also increased significantly in mice treated with Sunitinib compared to controls. Finally, increased engraftment in Sunitinib-treated samples was also seen along the length of the muscle (*Figure 12E*). Taken together, this shows that Sunitinib treatment improves the regenerative capacity of human FSHD myoblasts by increasing their engraftment and myogenic differentiation.

## Discussion

We report that Ret is a novel mediator of muscle stem cell function, is regulated by DUX4 and con-tributes to the pathogenic phenotype of FSHD. We focused on Ret signalling as a potential mediator of DUX4-induced muscle pathology for several reasons. FSHD is characterised by the often asym-metric wasting of facial, shoulder and upper arm muscles (*Tawil, 2008*), and can be associated with several non-muscle tissue pathologies including sensorineural hearing loss and retinal vasculopathies (*Verhagen et al., 1995*). Development of the auditory system, retinal patterning and facial muscles have been shown to depend on Ret signalling in a number of species. *Ret* is expressed in the second pharyngeal arch, from which some facial muscles arise, both in mouse and zebrafish (*Knight et al., 2011*; *Pachnis et al., 1993*). Additionally, Ret, GFRα1 and GFRα2 are expressed in the retina and the sensory epithelium of the cochlea, both of which can be affected in FSHD (*Golden et al., 1999*; *Zordan et al., 2006*; *Yu et al., 1998*). Phenotypic analysis of *Ret*, *GFRa* (*Enomoto et al., 1998*; *Tomac et al., 2000*; *Rossi et al., 1999*; *Honma et al., 2002*; *Lindahl et al., 2000*) and *GFL* (*Honma et al., 2002*; *Moore et al., 1996*; *Heuckeroth et al., 1999*; *Tomac et al., 2002*)-mutant mice have not reported a skeletal muscle phenotype, although this may have been overlooked due to the severe phenotype and perinatal lethality of some mutant lines. However, *Ret* is expressed in the indirect flight muscles of *Drosophila*, and overexpression of constitutively active *Ret* (RET$^{MEN2B}$) in muscle via a MEF2-GAL4 driver causes increased muscle fibre size, irregular myofibrils and abnor-mal actin deposits (*Klein et al., 2014*). Interestingly, irregular myotube morphology is often reported in FSHD patient-derived cultures (*Barro et al., 2010*; *Tassin et al., 2012*). As we found that Ret was activated by DUX4 in murine satellite cells, it becomes a potential candidate for contributing to DUX4-induced myopathy.

There has been no description of Ret function during myogenesis in mammals, despite the descriptions of Ret pathway genes expressed in specific muscle cell populations, including facial muscle precursors (*Golden et al., 1999*; *Natarajan et al., 2002*; *Knight et al., 2011*). We found that *Ret* and three of its co-receptors: *Gfrα1*, *Gfrα2* and *Gfrα4* are dynamically expressed in mammalian limb satellite cells. Intriguingly, *Ret* expression was higher during proliferation, whereas *Gfrα1, Gfrα2* and *Gfrα4* expression increased upon differentiation. This reciprocal expression pattern between *Ret* and its co-receptors also occurs in other cell types (*Li et al., 2009*) and so may regulate Ret signal-ling by feedback loops. Additionally, the ligands GDNF and PSPN, required to activate Gfrα1 and Gfrα4 (and to a lesser extent Gfrα2), are expressed by muscle fibres (*Golden et al., 1999*; *Yang and Nelson, 2004*; *Lindahl et al., 2000*), suggesting that myofibres may signal to satellite cells via Ret signalling. We also found that *RET* is expressed at low levels in primary human myoblasts, revealing a conserved expression in vertebrate muscle. Constitutive-expression and knockdown studies revealed that Ret signalling is required to maintain a proliferating myoblast population and prevent premature cell cycle exit and myogenic differentiation. Specifically, constitutively active RET

increased satellite cell proliferation but did not affect terminal differentiation. Consistent with this, siRNA-mediated knockdown of Ret reduced *Pax7* expression, concomitant with reduced satellite cell proliferation, whilst increasing differentiation. Knockdown of Gfrα1, Gfrα2 or Gfrα4 causes similar, if less severe, defects suggesting that Ret is interacting with its Gfrα co-receptors to control the switch from proliferation to differentiation in myoblasts.

Having shown that Ret is involved in the control of myogenesis, we aimed to understand its role in mediating DUX4 pathology. DUX4 is pathogenic in both myoblasts and differentiated muscle (*Vanderplanck et al., 2011*; *Bosnakovski et al., 2008a, 2014*; *Kowaljow et al., 2007*; *Wallace et al., 2011*; *Dandapat et al., 2014*). Recent microarray and ChIP-seq datasets reveal that DUX4 causes wide-ranging transcriptional disruption in myogenic cells (*Geng et al., 2012*; *Banerji et al., 2015*; *Rahimov et al., 2012*; *Knopp et al., 2016*) , with its potent transcriptional activation through its C-terminal domain (*Geng et al., 2012*; *Clapp et al., 2007*; *Kawamura-Saito et al., 2006*). However, identifying which disrupted pathways directly contribute to DUX4-specific pathology is crucial when considering future drug design to treat FSHD.

Ret is genetically downstream of DUX4 in myoblasts, but does not appear to be a direct transcriptional target. Transgenic analysis has shown that in mouse, 1.9 kb 5′ of the Transcriptional Start Site (TSS) of *Ret* is sufficient to drive tissue specific expression (*Zordan et al., 2006*). However, analysing two published ChIP-Seq datasets of DUX4 overexpression in human myoblasts (*Geng et al., 2012*; *Choi et al., 2016*), we failed to identify significant peaks within 15 kb 5′ of the *Ret* TSS. In addition, sequence analysis also failed to find a DUX4 consensus binding site (TAATCTAATCA – [*Zhang et al., 2016*]) within this same 15 kb region. It is of note however, that the C-terminus of DUX4 can recruit acetyltransferases (p300/CBP) to histones to induce acetylation, hence promoting transcription at sites distant to DUX4 DNA binding (*Choi et al., 2016*), so DUX4 could still directly controlling *Ret* in an epigenetic manner. However, also arguing against Ret being a direct transcriptional target of DUX4 is our observation that *Ret* expression is not only enhanced by DUX4 and the constitutively active tMALDUX4-VP16, but also by our DUX4 dominant-negative tMALDUX4-ERD construct in the microarray analysis. The tMALDUX4-ERD construct contains the two DNA binding homeodomains of DUX4, so should select the same cohort of transcriptional targets as DUX4. However, if DUX4 binds directly to the *Ret* promoter, tMALDUX4-ERD should suppress transcription and so reduce Ret expression, but this is the opposite to what we observe. Together, these observations indicate that DUX4 affects *Ret* expression indirectly, via binding to genes that, in turn, control *Ret* expression.

Blocking *Ret* expression or signalling increases the probability of DUX4-expressing myoblasts to differentiate into multinucleate myotubes. Despite this, some aspects of DUX4-mediated pathologies were not rescued by Ret inhibition. For example, inhibition of Ret using Sunitinib did not alter DUX4-induced apoptosis. DUX4 transcriptionally dysregulates a large number of pathways, so it is unlikely that one pathway only is responsible for all pathological characteristics (*Banerji et al., 2015*). For example, we have recently shown that β-catenin signalling is important for DUX4-mediated transcriptional dysregulation in skeletal muscle (*Banerji et al., 2015*).

To investigate the relative importance of Ret in mediating DUX4 action, we adopted a pharmacological approach that allowed us to modify Ret signalling in a dose-dependent manner. As the readout, we used the suppression of differentiation caused by constitutive CA RET51 (RET51-MENA) expression in murine C2C12 myoblasts and asked at which doses we could inhibit Ret to rescue differentiation, without causing deleterious side-effects. Comparison of Ret inhibitors Zactima (Vandetanib/ZD6474) (*Carlomagno et al., 2002*), TG101209 (*Pardanani et al., 2007*) and Sunitinib (Sutent, SU11248) (*Kim et al., 2006*) revealed that Suntinib showed the most robust rescue of CA RET51-mediated inhibition of myogenesis. We further tested the ability of Sunitinib to rescue DUX4-mediated phenotypes in murine satellite cells and showed that we could again rescue differentiation, but not proliferation: consistent with the effects of siRNA-mediated knockdown of *Ret*. Zactima, TG101209 and Sunitinib also inhibit other receptor tyrosine kinases, including *VEGFR*, *EGF* and *MET* (*Plaza-Menacho et al., 2014*). Of note, *Vegfr2* is also a DUX4 target identified in our microarray screen (*Banerji et al., 2015*), highlighting another potential mediator of DUX4 pathology in myoblasts. Since many of these receptors activate the same intracellular pathways as Ret, it will be important to understand their relative roles and redundancies, when activated in the context of FSHD.

Since DUX4 up-regulates *Ret,* and Sunitinib can clearly rescue the differentiation block caused by CA-RET51 in C2C12 myoblasts, we investigated intracellular signalling events. ERK and AKT are two key intracellular mediators of Ret signalling e.g. (*Mograbi et al., 2001*). ERK1/2 phosphorylation was inhibited by Sunitinib in DUX4-expressing iC2C12-DUX4 myoblasts, but not in control myoblasts not expressing DUX4, suggesting a potential mechanism by which DUX4-mediated Ret signalling could inhibit differentiation. In contrast, AKT phosphorylation was unaltered by inducing DUX4. Although Ret can operate through ERK1/2 signalling, it is possible that Sunitinib could also be acting on ERK1/2 phosphorylation activated by other pathways perturbed by, or in response to, DUX4.

To assess whether Sunitinib could be used to ameliorate FSHD pathologies in man, we used human myoblast lines isolated from a mosaic FSHD1 patient (*Krom et al., 2012*). We assessed whether Sunitinib could rescue cell morphology, proliferation and the differentiation capacity of human myoblasts displaying FSHD pathologies. Using statistical models we showed that Sunitinib could induce the FSHD myoblasts to adopt the phenotype of isogenic myoblasts lacking the D4Z4 contraction. Transplantation assays of FSHD myoblasts into cryodamaged immunodeficient mice showed increased engraftment of FSHD myoblasts occurred in response to daily systemic Sunitinib treatment. Therefore, this pre-clinical model for evaluating modifiers of FSHD pathology identifies Sunitinib as a potential therapeutic agent for FSHD.

In conclusion, we have shown that Ret is a novel regulator of satellite cell function in mammalian muscle, and contributes to the FSHD phenotype. Specifically, blocking DUX4-mediated Ret signalling increases myogenic differentiation, giving further insight into the molecular pathology of DUX4 and highlighting Ret signalling as potential drug target. Crucially, the clinically-approved RET inhibitor Sunitinib (*Widmer et al., 2014*) rescues cell morphology, proliferation and myogenic differentiation in human FSHD myoblasts both in vitro and in vivo. Given that FSHD is currently incurable, our findings highlight RTK inhibitors as a potential novel therapeutic strategy to treat muscle wasting in FSHD.

## Materials and methods

### Animals

Experimental procedures were performed in accordance with British law under the provisions of the Animals (Scientific Procedures) Act 1986, as approved by the King's College London and University College London Ethical Review Process committees. C57BL/10 male mice aged between 8 and 10 weeks were used for myofibre and satellite cell-derived myoblast preparations. Immunodeficient NOD/scid/γ-chain mice were used for grafting experiments, performed under UK Home Office Project Licence 70/8566. All surgery was performed under isoflurane anesthesia, and every effort made to minimize pain and suffering, including use of analgesics.

### Cell culture and primary satellite cell preparation

Single myofibres were isolated from the extensor digitorum longus (EDL) as previously described (*Moyle and Zammit, 2014*). Briefly, dissected EDLs were digested in DMEM (+Glutamax) (Thermo Fisher Scientific, Waltham, MA) containing 0.2% collagenase (Sigma-Aldrich, Dorset, UK) for 2 hr in a 37°C 5% $CO_2$ incubator before manual disruption with a heat-polished glass pipette in 5% bovine serum albumin (BSA) (Sigma Aldrich) coated dishes. Individual washed myofibres with associated quiescent satellite cells were subsequently fixed in 4% paraformaldehyde/PBS (PFA) or cultured in non-adherent or adherent conditions. Non-adherent cultures were grown in DMEM-Glutamax (Thermo Fisher Scientific) supplemented with 10% horse serum (HS) (Thermo Fisher Scientific), 0.5% chicken embryo extract (CEE) and 1% Penicillin-Streptomycin (Sigma Aldrich) for up to 72 hr. For adherent cultures of proliferating satellite cells, myofibres were plated on 1 mg/ml Matrigel (Collaborative Research Inc., Bedford, MA) coated dishes in DMEM (Glutamax) supplemented with 20% foetal calf serum (FBS)-Gold (Thermo Fisher Scientific), 10% HS, 1% CEE, 10 ng/ml basic FGF (Peprotech, Rocky hill, NJ) and 1% Penicillin-Streptomycin for 72 hr. Subsequently, myofibres were removed by pipette agitation and satellite cells re-plated after trypsinisation in 0.25% Trypsin-EDTA.

Primary human myoblasts were obtained from biopsies of the vastus lateralis of consenting individuals (approved by the UK National Health Service Ethics Committee (London Research Ethics Committee; reference 10/H0718/10 and in accordance with the Human Tissue Act and Declaration

of Helsinki). Biopsies were digested in basal medium (PromoCell containing collagenase B and dispase II) and single cells isolated via 100 µM cell strainer as previously described (*Agley et al., 2015*). Adherent cells were cultured for 7 days in skeletal muscle cell growth medium (PromoCell, Heidelberg, Germany) and the NCAM/CD56+ myogenic population collected via magnetic activated cell sorting (MACS).

The immortalised human myoblast lines 54.6 (un-contracted *D4Z4*, no DUX4 expression) and 54.12 (contracted *D4Z4*, DUX4 expression) from a mosaic patient were a kind gift from V. Mouly and Silvere van der Maarel (*Krom et al., 2012*) and were verified by assessing *DUX4* expression and tested for mycoplasma. Both lines were maintained in skeletal muscle cell growth medium (PromoCell, C-23160) and differentiated in skeletal muscle cell differentiation medium (PromoCell supplemented with 50 µg/ml Gentacmycin).

C2C12 myoblasts (*Yaffe and Saxel, 1977*), iC2C12-DUX4 and iC2C12-DUX4c myoblasts (*Bosnakovski et al., 2008a*) and HEK 293T cells were maintained in DMEM supplemented with 10% FBS, 1% L-Glutamine (Sigma Aldrich) and 1% Penicillin-Streptomycin. For differentiation, both satellite cell-derived myoblasts and C2C12 myoblasts were cultured in mitogen-poor medium, containing DMEM, 2% HS, 1% Penicillin-Streptomycin and 1% L-Glutamine. iC2C12-DUX4 and iC2C12-DUX4c myoblasts were tested for DUX4 and DUX4c induction respectively.

## Retroviral expression constructs

Plasmids were obtained encoding human *RET9* and *RET51* and constitutively active (CA) forms containing the Cys634Lys mutation that occurs in multiple endocrine neoplasia type 2A (MEN2A) patients (*de Graaff et al., 2001*). Coding sequences were sub-cloned into a modified *pMSCV-puro* vector (Takara Bio Europe, Saint-Germain-en-Laye, France), in which the puromycin resistance gene was replaced with an internal ribosomal entry site (IRES) and enhanced green fluorescent protein (eGFP), allowing transduced cells to be identified due to the presence of eGFP (*Zammit et al., 2006*). The DUX4 and DUX4c constructs have previously been published (*Banerji et al., 2015*). All constructs were sequenced to ensure fidelity.

Retroviruses were produced using HEK 293T packaging cells, by co-transfecting the expression vectors in the presence of an ectopic helper plasmid. Replication incompetent viral particles were harvested from the culture medium and expression confirmed by western blot and immunofluorescence.

## Retroviral transduction

C2C12 myoblasts and satellite cell-derived myoblasts were plated at 70% confluency and transduced with control, *RET, RET* mutants, *DUX4c* or *DUX4*-encoding retroviruses in the presence of 4 µg/ml Polybrene for 6 hr at 37°C, 5% $CO_2$ in proliferation medium, before the medium was replaced.

## siRNA-mediated gene knockdown

Satellite cells were transfected with 20 nM of control or *Ret* Silencer Select siRNA (Thermo Fisher Scientific) in the presence of Lipofectamine RNAiMAX (Thermo Fisher Scientific) for 6 hr at 37°C, 5% $CO_2$ in proliferation medium. The siRNA sequence 5'-GCUUGUACAUCGGGACUUATT-3' (ID: s72895) was used to knockdown murine *Ret* expression, and control siRNA was supplied by Thermo Fisher Scientific. Gene knockdown was confirmed 48 hr after transfection.

## Ret signalling inhibition using clinically approved drugs

Three small molecule tyrosine kinase inhibitors were analysed for their ability to inhibit RET signalling. All drugs are ATP-competitive inhibitors of tyrosine kinases, and were obtained from Selleckchem.com: Sunitinib (Sutent, SU11248) (*Kim et al., 2006*), TG101209 (*Pardanani et al., 2007*) and Zactima (*Carlomagno et al., 2002*). The chemical structures of each compound can be found in (*Pardanani et al., 2007*; *Kim et al., 2006*; *Wedge et al., 2002*).

## Quantitative RT-PCR

Cells were cultured in 6-well plates for at least 48 hr under experimental conditions and total RNA extracted using RNeasy Kit (QIAGEN Ltd, Manchester, UK ). Between 500 ng – 1 µg of RNA was used to prepare cDNA using the QuantiTect Reverse Transcription Kit with genomic DNA wipeout

(QIAGEN). RT-qPCR was performed on an Mx3005PQPCR system (Agilent Technologies LDA UK Ltd, Stockport, UK) with MESA Blue qPCR MasterMix Plus and ROX reference dye (Eurogentec Ltd, Southampton, UK). Primers used were as follows:

Total *Ret* (F: 5'-AAGCAGGAGCCAGACAAGAG-3' and R: 5'-ACACCTTCGGACTCACTGCT-3'),

*Ret9* (F: 5'-GATCCAGAGGCCAGACAAC -3' and R: 5'- GTAGAATCTAGTAAATGCA-3'),

*Ret51* (F: 5'-GATCCAGAGGCCAGACAAC-3' and R: 5'-AGGACTCTCTCCAGGCCAG-3'),

*GFRα1* (F: 5'-GCACAGCTACGGGATGCTC-3' and R: 5'-CTCTGGCTGGCAGTTGGT-3'),

*GFRα2* (F: 5'-ACCGTGTGCCCAGCGAGTATA-3' and R: 5'-CGACAGTTGGCGTGGAAGT-3'),

*GFRα3 (F: 5'- GGAAAATGAATCTTAGCAAGTTGAA-3'* and R: 5'- TTGTCGTGAAGAGTACACAGCA TAG-3'),

*GFRα4* (F: 5'-ACCCCTGCTTGGATGGTGCC-3' and R: 5'-CAGCCAGGACACCTTGGGCG-3'),

*Gapdh* (F: 5'-GTGAAGGTCGGTGTGAACG-3' and R: 5'-ATTTGATGTTAGTGGGGTCTCG-3')

*Tbp* (F: 5'-ATCCCAAGCGATTTGCTG-3' and R: 5'-CCTGTGCACACCATTTTTCC-3')

Primers for murine *Pax7*, *MyoD* and *Myf5* have previously been published (*Collins et al., 2009*). Human primers were as follows:

*RPLPO* (F: 5'- TCTACAACCCTGAAGTGCTTGAT-3' and R: 5'- CAATCTGCAGACAGACACTGG-3')

*Cyclin D1* (F: 5'-GCCGAGAAGCTGTGCATC -3' and R: 5'-CCACTTGAGCTTGTTCACCA-3'),

*MYOD* (F: 5'- GCTCCGACGGCATGATGG -3' and R: 5'- GACACCGCCGCACTCTTCCC -3'),

*MYOG* (F: 5'- CCAGGGGTGCCCAGCGAATG -3' and R: 5'- AGCCGTGAGCAGATGATCC -3'),

*MyHC* (F: 5'- AGCAGGAGGAGTACAAGAAG -3' and R: 5'- CTTTGACCACCTTGGGCTTC -3') and

*RET* (F: 5'-GCTCCACTTCAACGTGTC-3' and R: 5'-GCAGCTTGTACTGGACGTT-3').

Murine samples were normalised to the housekeeping genes *Gapdh* and/or *Tbp* (as designated) and human samples normalised to *RPLPO*.

## Immunolabelling

Floating myofibres or plated satellite cells were fixed in 4% paraformaldehyde/PBS, permeabilised with 0.5% Triton/PBS (Sigma Aldrich) and blocked in PBS containing 5% swine serum + 5% goat serum (DakoCytomation, Glostrup, Denmark) for 1 hr (except for samples using goat RET51 antibody, which were blocked in 10% swine serum). Samples were incubated in primary antibody overnight at 4°C, washed 3 times in 0.025% Tween/PBS and visualised by incubating with AlexaFluor conjugated secondary antibodies (Thermo Fisher Scientific) at 1/500 dilution for 1 hr at room temperature. Nuclei were visualised by mounting in aqueous mountant containing DAPI (Vectashield), for further detail see (*Moyle and Zammit, 2014*). Primary antibodies used were: goat anti-RET (C-20, Santa Cruz Biotechnology, Dallas TX, 1/150), rabbit anti-RET51 (C-19, Santa Cruz Biotechnology, 1/150) and rabbit anti-pRET (Tyr1062-R, Santa Cruz Biotechnology, 1/150), mouse anti-Pax7 (Developmental Studies Hybridoma Bank, Iowa City, IA, 1/10), mouse anti-Myogenin (F5D, Developmental Studies Hybridoma Bank, 1/10) and mouse anti-MyHC (MF-20, Developmental Studies Hybridoma Bank, 1/300), rabbit anti-eGFP (A11122, Thermo Fisher Scientific, 1/500), chicken anti-eGFP (ab13970, Abcam, Cambridge, UK, 1/1000), rabbit anti-phospho-Histone H1 (06–597 Thermo Fisher Scientific, 1/300), rabbit anti-phospho-Histone H3 (06–570, Thermo Fisher Scientific, 1/100), rabbit anti-Desmin (D93F5, Cell Signalling Technology, Danvers, MA, 1/250), rabbit anti-Ki67 (SP6, A. Menarini Diagnostics Ltd, Winnersh, UK, 1/200) and mouse anti-tubulin (E7-C, Developmental Studies Hybridoma Bank, 1/1000).

EdU analysis was performed with the Click-iT EdU Alexa Fluor Imaging Kit, by incubating cells for 2 hr with 10 μM of 5-Ethynyl-2'-deoxyuridine (EdU) before fixation and processing as per manufacturer's instructions (Thermo Fisher Scientific).

## Immunoblot analysis

HEK 293T cells were transfected with RET expression vectors for 24 hr and total protein extracted in the presence of a complete protease inhibitor cocktail (PIC, Roche Diagnostics Ltd, Burgess Hill, UK). Samples were quantified using the DC Protein Assay (Biorad Laboratories, Hercules, CA, 500–0116) and equal quantifies run on pre-cast 4–20% electrophoresis gels (Thermo Fisher Scientific) with 0.35 μl dithiothreitol and bromophenol blue dye at 120 V. Gels were transferred to PVDF membranes using the iBlot dry-blotting system (Thermo Fisher Scientific), blocked in 5% non-fat milk

powder/PBS (Marvel: Premier Foods, St Albans, UK) and incubated overnight at 4°C in primary anti-body dissolved in 1% milk powder/PBS.

iC2C12-DUX4 myoblasts (*Bosnakovski et al., 2008a*) were plated at 3000 cells/cm$^2$ and DUX4 expression induced by the addition of 500 ng/ml doxycycline hyclate (Thermo Fisher Scientific, J60579) with/without 250 ng/ml Sunitinib for 20 hr. Myoblasts were lysed in RIPA buffer (CST, 9806S) and subjected to three freeze/thaw cycles and the lysate incubated on ice for 20 min and then centrifuged for 20 min at 4°C. Protein concentration was quantified using the DC Protein Assay (Biorad, 500–0116), and 40 μg protein was mixed with 5x Laemmli Buffer, subjected to 10% SDS-PAGE and transferred onto a nitrocellulose membrane (Sigma-Aldrich, WHA10402578). Membranes were blocked in 5% non-fat milk powder in TBS/0.1% Tween 20 (TBS/T) (Sigma-Aldrich, P5927) and incubated with primary antibodies overnight at 4°C. Primary antibodies used were goat anti-RET (C20, Santa Cruz Biotechnology, 1/150), rabbit anti-RET51 (C-19, Santa Cruz Biotechnology, 1/150), rabbit anti-phospho-p44/42 MAPK (Erk1/2) (Thr202/Tyr204) (9101, Cell Signalling Technology, 1/1000), rabbit anti-total p44/42 MAPK (Erk1/2) (137F5, Cell Signalling Technology, 1/1000), rabbit anti-phospho-Akt (Ser473) (D9E, Cell Signalling Technology 1/2000), mouse anti-total AKT (40D4, Cell Signalling Technology, 1:2000), mouse anti-DUX4 (9A12, Merck Millipore, 1/1000) and rabbit anti-Caveolin-1 (sc-894, Santa Cruz Biotechnology, 1:500).

Membranes were then washed and protein bands visualised by incubating with horseradish per-oxidase (HRP)-conjugated secondary antibodies (GE Healthcare, Chalfont St Giles, UK, Na934V, NA931V, 1/5000) for 1 hr at room temperature. Membrane bound protein was visualised using clar-ity western ECL substrate (BioRad, 170–5061), and band intensity was quantified with ChemiDoc MP System (BioRad). For iC2C12-DUX4 myoblasts, band intensities were normalised to housekeeping protein Caveolin-1. The ratios of pERK:tERK and pAKT:tAKT in the treated groups were subse-quently compared to the ratios in the control group.

## Apoptosis assay and FACs

iC2C12-DUX4 myoblasts were induced to express DUX4 with 250 ng/ml or 500 ng/ml doxycycline for 24 hr in the presence of 0, 250 ng/ml or 500 ng/ml Sunitinib malate. Samples were trypsinised and stained with APC-conjugated Annexin V Apoptosis detection kit, according to manufacturer's instructions (BD Biosciences, Franklin Lakes, NJ). Samples were run on a CyAn ADP flow cytometer (Beckman Coulter) and analysed using FlowJo software.

## Human myoblast transplantation

Human cell work in the Tedesco laboratory has been conducted under the approval of the National Health Service Health Research Authority Research Ethics Committee, reference no. 13/LO/1826; IRAS project ID no. 141100. Transplants were performed under UK Home Office Project Licence 70/8566. Immortalised human FSHD clone 54.12 myoblasts were cultured at low density in skeletal mus-cle cell growth medium (Promocell, C-23160) and separate cohorts either pre-treated 24 hr before transplantation with 250 ng/ml Sunitinib malate or control medium. NOD/scid/γ-chain knockout immunodeficient mice were given an intraperitoneal injection of 20 mg/kg/day Sunitinib malate dis-solved in Ca$^{2+}$/Mg$^{2+}$-free PBS (n = 4), while non pre-treated controls received PBS (n = 4). 24 hr later, TA muscles of all eight NOD/scid/γ-chain knockout immunodeficient mice were cryoinjured with a metal probe pre-cooled in liquid nitrogen for 30 s to induce regeneration. TA muscles of the four mice pre-treated with Sunitinib were then injected using a 30G syringe (BD Biosciences) with 1 × 10$^6$ 54.12 FSHD myoblasts that had been pre-treated with Sunitinib in vitro, while 1 × 10$^6$ non-treated 54.12 FSHD myoblasts per mouse were injected into PBS-treated mice (n = 4) (*Gerli et al., 2014*). Mice grafted with myoblasts pre-treated with Sunitinib then received daily intraperitoneal injections of 20 mg/kg/day Sunitinib malate dissolved in Ca$^{2+}$/Mg$^{2+}$-free PBS, while non pre-treated controls received PBS. Three weeks after transplantation, mice were humanely killed and grafted TA muscles removed and frozen in isopentane cooled in liquid nitrogen. Muscles were mounted using gum tragacanth on cork and serial transverse sections were cut at 8 μm using a Leica CM 1850 UV cryostat (Leica Biosystems, Milton Keynes, UK).

## Analysis of cell engraftment

Cryosections 8 µm thick and spaced 112 µm along the length of the muscle were collected for analysis with human-specific antibodies to measure engraftment of donor myoblasts. Frozen sections were fixed in 4% PFA for 10 min, permeabilised with 0.1% Triton X100/1% BSA (Sigma Aldrich)/PBS and blocked in 10% donkey serum (Dakocytomation) in 0.1% Triton X100/1% BSA/PBS. Cryosections were then incubated with mouse anti-human LAMIN A/C (NCL-LAM, Leica Biosystems) and mouse anti-human SPECTRIN (NCL-SPEC1, Leica Biosystems) to locate engrafted cells and myofibres, together with rabbit anti-mouse Laminin (L9393, Sigma Aldrich) to immunostain extracellular matrix and delimit myofibres. Samples were counterstained with Hoechst to identify total nuclei. The total number of donor-derived LAMIN A/C-positive nuclei within or outside of myofibres, and donor-derived SPECTRIN-positive myofibres were counted in each section along the grafted TA and expressed either as a total per mouse (all sections combined) or an average per section. Quantification was performed blinded with the experimenter unaware of whether the cells were from Sunitinib-treated or control mice.

## Statistics

Data was analysed using either two-tailed Student's $t$-tests, or by statistical models fitted to the data that incorporated random effects due to mouse or between experiment variations (as determined in text). Quasi-Poisson models were used for evaluating the significance of differences in the numbers of cells between conditions and mixed binomial models employed likewise for ratios including the fusion index. Models tested either the linear relationship between a factor and the result, or each combination of factors as an independent parameter. The fusion index was expressed as the ratio of the number of nuclei incorporated into MyHC+ fibres divided by the total number of nuclei in the field of view. Statistical models and analyses were generated using the R programming language (*R Core Team, 2011*) and Excel.

## Image analysis

Cell eccentricity analysis was performed on Tubulin immunolabelled cells using a customised R script, written using the EBImage package (*Pau et al., 2010*) . Images were first pre-processed using a low pass filter to remove low intensity background staining and a high pass filter to exclude non-specific high intensity background based on size and morphology. Cells with overlapping cytoplasm were segregated into independent objects prior to quantification using Voroni segmentation in which DAPI nuclear staining was used as an anchor to define cells; annotated images of segregation were output for visual assessment of correct segregation. The eccentricities of each cell cytoplasm (defined by Tubulin immunolabelling in pre-processed, segregated images) was then computed for each image and the average eccentricity of the frame was obtained, eccentricity computes the cell shape deviation from a circle (1) to a line (0) (Source Code file-Measuring Cell Eccentricity.r).

## Acknowledgements

LAM was supported by the Muscular Dystrophy UK [grant number RA4/817 to PZ], with additional support from a Medical Research Council Doctoral Training Grant awarded to King's College London and by the Association Française contre les Myopathies (grant numbers 15814 and 19105 to PZ and RK). BBSRC support for this work was through grants to SH and OJ (BB/L009943/1), RK (BB/D020433/1) and RK, SH and PZ (BB/I025883/1). The Zammit laboratory is additionally supported by the Medical Research Council, European Union's Seventh Framework Programme for research, technological development and demonstration under grant agreement number 262948–2 (BIODESIGN) and the FSH Society Shack Family and Friends research grant (FSHS-82013-06). The Tedesco laboratory is supported by the European Union's Seventh Framework Programme for research, technological development and demonstration under grant agreement no. 602423 (PluriMes), the UK National Institute for Health Research (NIHR), the BBSRC, Duchenne Parent Project Onlus, Fundació la Marató de TV3 and Muscular Dystrophy UK.

Constructs for human and mouse RET were a kind gift from Tiffany Heanue and Vasilis Pachnis at NIMR, Mill Hill. Mosaic FSHD myoblast lines (*Krom et al., 2012*) were a kind gift from Vincent Mouly and Silvere van der Maarel and iC2C12-DUX4 and iC2C12-DUX4c myoblasts (*Bosnakovski et al.,*

*2008a*) a kind gift from Michael Kyba. We would like to thank Huascar Ortuste Quiroga and Livia Katonova for processing mouse tissue.

---

## Additional information

### Funding

| Funder | Grant reference number | Author |
|---|---|---|
| Muscular Dystrophy UK | RA4/817 | Louise A Moyle<br>Peter S Zammit |
| Biotechnology and Biological Sciences Research Council | BB/L009943/1 | Oihane Jaka<br>Stephen DR Harridge |
| European Union's Seventh Framework Programme for research, technological development and demonstration | 602423 | Francesco Saverio Tedesco |
| Biotechnology and Biological Sciences Research Council | BB/I025883/1 | Stephen DR Harridge<br>Robert D Knight<br>Peter S Zammit |
| Biotechnology and Biological Sciences Research Council | BB/L009943/1 | Stephen DR Harridge |
| Biotechnology and Biological Sciences Research Council | BB/D020433/1 | Robert D Knight |
| Association Française contre les Myopathies | 19105 | Robert D Knight<br>Peter S Zammit |
| FSH Society | FSHS-82013-06 | Peter S Zammit |
| Association Française contre les Myopathies | 15814 | Peter S Zammit |
| European Union's Seventh Framework Programme for research, technological development and demonstration | 262948-2 | Peter S Zammit |

The funders had no role in study design, data collection and interpretation, or the decision to submit the work for publication.

### Author contributions

LAM, RDK, Conception and design, Acquisition of data, Analysis and interpretation of data, Drafting or revising the article; EB, Conception and design, Analysis and interpretation of data; OJ, SDRH, Acquisition of data, Analysis and interpretation of data; JP, Helped design, perform and analyse the Western Blots to examine the effects of Sunitinib on ERK and AKT mediated signalling in response to DUX4, and presented as Figures 7H-J in the revised manuscript; CRSB, Wrote software to analyse cell eccentricity, Acquisition of data, Analysis and interpretation of data; FST, Helped design, perform and analyse the grafting experiments to test the effects of Sunitnib in vivo and presented as Figure 12 in the revised manuscript; PSZ, Conception and design, Analysis and interpretation of data, Drafting and revising the article

### Author ORCIDs

Francesco Saverio Tedesco, http://orcid.org/0000-0001-5321-7682
Robert D Knight, http://orcid.org/0000-0001-9920-836X
Peter S Zammit, http://orcid.org/0000-0001-9562-3072

### Ethics

Human subjects: Primary human myoblasts were obtained from biopsies of the vastus lateralis of consenting individuals (approved by the UK National Health Service Ethics Committee (London Research Ethics Committee; reference 10/H0718/10 and in accordance with the Human Tissue Act and Declaration of Helsinki).

Animal experimentation: Experimental procedures were performed in accordance with British law under the provisions of the Animals (Scientific Procedures) Act 1986, as approved by the King's College London and University College London Ethical Review Process committees. Immunodeficient NOD/scid/γ-chain mice were used for grafting experiments, performed under UK Home Office Project Licence 70/8566. All surgery was performed under isoflurane anesthesia, and every effort made to minimize pain and suffering, including use of analgesics. Human cell work in the Tedesco lab has been conducted under the approval of the National Health Service Health Research Authority Research Ethics Committee, reference no. 13/LO/1826; IRAS project ID no. 141100.

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
