## [Decision Letter]

Thank you for submitting your work entitled "Discovery of a novel role for Ret tyrosine kinase in muscle stem cells provides a platform for therapy in FSHD" for consideration by *eLife*. Your article has been reviewed by three peer reviewers, and the evaluation has been overseen by Janet Rossant as the Senior Editor and Reviewing Editor.

The reviewers have discussed the reviews with one another and the Reviewing editor has drafted this decision to help you prepare a revised submission.

Summary:

Moyle et al. describe the induction of receptor tyrosine kinase Ret in mouse myoblast cultures following expression of the FSHD causal gene DUX4. The authors investigate the expression pattern of Ret and its co-receptors GRFα1,2,4 during normal myoblast differentiation. They perform loss of function studies showing Ret involvement in myoblast proliferation activation and differentiation inhibition. They further show that Ret down regulation (either by mRNA silencing or kinase inhibitory drugs) rescues the myogenic differentiation defects induced by DUX4.

The study is original, is supported by appropriate experiments and elegantly shows a putative new therapeutic strategy for FSHD with already approved drugs such as Sunitinib. This research is a first step for the identification of therapeutic targets for human FSHD patients.

The reviewers had an extensive discussion and agreed on the potential of this work to provide novel therapy for FSHD. Given the importance of this finding, they strongly felt that there were several key experimental issues that needed to be addressed to ensure the validity of the findings, particularly in terms of in vivo relevance. Listed below are the major experimental issues that they would like to see addressed. In addition, all reviewers expressed concern with the presentation of the data and the statistical analysis. We have tried to define a set of experiments that are achievable in a reasonable time frame and we look forward to receiving a revised manuscript.

Essential revisions:

1) There hasn't been clear evidence that Ret is expressed in mouse muscle progenitors and the papers cited both in the Introduction and Discussion do not support the claim (except for the zebrafish paper Knight et al., 2011). It is essential that authors provide clear evidence that Ret is expressed in vivo at some stage of muscle development and/or homeostasis.

2) The lack of an animal model for FSHD or DUX4 OE is not a sufficient reason to avoid in vivo experiments. To really test the physiological effect of Ret inhibition or Sunitinib treatment, the authors can use myoblast transfer experiments to assess the in vivo rescue of DUX4 OE cells or FSHD patient cells. These experiments are technically straight-forward and have been used for 10+ years. The authors could trace injected cells by GFP and measure the contribution to muscle regeneration of wild type, DUX4 OE, DUX4 OE treated with siRNA against Ret, and DUX4 OE treated with Sunitinib. The authors can use human myoblasts to really provide the disease relevance of their proposed therapy. These experiments will greatly enhance the significance and impact of this study.

3) In Figure 5, DUX4 OE experiments were used to suggest that Ret is a DUX4 target gene. The authors suggest that Ret is an indirect target that is not directly bound by DUX4 homeodomain. However, the expression of the tMALDUX4-VP16 construct did lead to a increase in Ret expression in Figure 5. In this construct, VP16 is a strong transcriptional activator and should trigger genes near DUX4 binding sites. Thus, it should be considered that Ret could still be a direct DUX4 homeodomain-binding target. The differences between DUX4 and DUX4c are not well defined and the authors should not interpret their results as such. If they do intend to show that Ret is or isn't a direct target of DUX4 DNA-binding, luciferase expression vectors with the proximal promoters of Ret should be used in combination with DUX4 OE to explicitly address this question.

4) Furthermore, the dominant negative tMALDUX4-ERD construct also increased Ret expression but was not discussed or mentioned in the text. The use of the Engrailed repressor domain should have shut down the DUX4 pathway, yet Ret is still upregulated. The authors should discuss this phenomenon. Is this the reason why the authors do not think that Ret is a direct target of DUX4?

5) Caspase 3/7 is not an accurate read out of cell death in the myogenic system. Previous studies have suggested the participation of capspases in the differentiation program. So the proportion of caspase 3/7 staining is affected by changes in differentiation. The authors should perform alternative measurements of cell viability (alamar blue, propidium iodide, etc.).

6) The authors take a candidate approach to identify RTK inhibitors using phenotypic readouts of cell proliferation and differentiation. However, to demonstrate the pharmacological inhibition of Ret, the authors should provide a read out of Ret inhibition by Sunitinib and other small molecules using a downstream signalling effector.

7) There seems to be a difference in the function of Ret in wild type myoblasts compared to its function in DUX4 OE or FSHD cells. Knockdown experiments in wild type cells decreased their proliferation, whereas Ret OE led to higher myoblast proliferation. However, in DUX4 OE and FSHD cells Ret inhibition through siRNA knockdown or Sunitinib treatment, proliferation of myoblasts did not change accordingly. The authors should discuss this difference. Are differences in the basal rates of proliferation in DUX4 OE and FSHD cells affected? Do these changes match the predicted change with Ret OE? These differences should be discussed further in the manuscript.

8) Statistics should be revised throughout the manuscript. Authors are frequently using paired t-test approach (e.g., Figure 1) although there are no paired data. They should consider either simple t-test or even one way ANOVA followed by multiple t-tests. Significance asterisks seem to be random as they are presented for only a subset of bars within graphs although clearly other bars are (appear to be) also significantly different from the controls (e.g.,, Figure 4).

[Editors' note: further revisions were requested prior to acceptance, as described below.]

Thank you for resubmitting your work entitled "Ret function in muscle stem cells points to tyrosine kinase inhibitor therapy for Facioscapulohumeral muscular dystrophy" for further consideration at *eLife*. Your revised article has been favorably evaluated by Janet Rossant and three reviewers.

The authors have correctly addressed most of the reviewers' critiques and have accordingly improved the clarity of their manuscript. The data presented about an unexpected pathway contributing to DUX4 pathological activity are really novel and exciting. RET / ERK involvement in FSHD and the possibility of targeting their pathway by a drug approved in cancer will most probably open new perspectives in the therapeutic approaches of FSHD. We are glad to see that the authors were able to perform myoblast engraftment assays with the human FSHD samples and demonstrate the efficacy of Sunitinib in vivo. However, there are still some mechanistic concerns, especially in regard to the action of Sunitinib.

These concerns can be addressed by modifying some of the claims of the paper, without new experiments.

1) A request for clarification at the end of the Discussion: If indeed ERK induces proliferation and inhibits differentiation, ERK inhibition by Sunitinib in DUX4-expressing cells is expected to do the opposite i.e. decrease proliferation besides increasing differentiation. Why is that decreased proliferation presented as an unexpected observation? What are the Authors suggesting as "another mechanism"?

2) While ChIP qPCR is a method to determine if DUX4 is a direct activator of Ret, the experiment lacks proper controls. There is no positive control to confirm that the ChIP experiment is working. There is also the possibility that the predicted site does not contain the DUX4 binding site.

Moreover, there is a rather glancing discussion on why the tMALDUX4-ERD construct also induced Ret expression. The Discussion will benefit from more critical perspectives of the results.

3) The identification of pERK1/2 as a readout of Ret activity is also weakly supported. pERK1/2 is not upregulated with DUX4 overexpression, therefore is not likely a Ret target. The decrease in ERK1/2 with Sunitinib treatment yields more questions than answers. It would suggest that Sunitinib is acting on ERK phosphorylation and not Ret activity.

While it can be argued that Sunitinib only inhibits ERK when DUX4 is expressed, it could also be due to the co-treatment with Doxycyclin.

---

## [Author Response]

*1) There hasn't been clear evidence that Ret is expressed in mouse muscle progenitors and the papers cited both in the Introduction and Discussion do not support the claim (except for the zebrafish paper Knight et al., 2011). It is essential that authors provide clear evidence that Ret is expressed* in vivo *at some stage of muscle development and/or homeostasis.*

We show that Ret is expressed by immunostaining in activated murine satellite cells associated with myofibres ex vivo (Figure 1). That gene expression is modulated in expanded satellite cells during myogenic differentiation using by RT-qPCR (Figure 1). Furthermore, RET can be detected by RT-qPCR in human satellite cell-derived primary myoblasts (Figure 2) and C2 cells (Figure 6).

We have tried using the commercially available antibodies on regenerating mouse muscle sections and embryos, but the immunostaining is not convincing, in contrast to immunolabeling cells ex-vivo. We have augmented the supporting literature showing Ret in muscle in the Introduction and Discussion and hope that we have now provided sufficient data/supporting evidence.

*2) The lack of an animal model for FSHD or DUX4 OE is not a sufficient reason to avoid* in vivo *experiments. To really test the physiological effect of Ret inhibition or Sunitinib treatment, the authors can use myoblast transfer experiments to assess the* in vivo *rescue of DUX4 OE cells or FSHD patient cells. These experiments are technically straight-forward and have been used for 10+ years. The authors could trace injected cells by GFP and measure the contribution to muscle regeneration of wild type, DUX4 OE, DUX4 OE treated with siRNA against Ret, and DUX4 OE treated with Sunitinib. The authors can use human myoblasts to really provide the disease relevance of their proposed therapy. These experiments will greatly enhance the significance and impact of this study.*

We had considered testing Sunitinib in an in vivo model for FSHD but it is not a particularly straightforward experiment, partly because of the lack of a suitable model (Lek, A., et al., Emerging preclinical animal models for FSHD. Trends Mol Med, 2015. 21(5): p. 295-306).

We therefore decided to use the immortalized human FSHD myoblast line 54.12 for grafting as this cell line has previously been shown to engraft into mouse muscle. Using this cell line we have been able to show that Sunitinib could improve proliferation/differentiation in vitro (Figure 10–Figure 11). We tried two models, transplantation into Zebrafish and into immuno-compromised mice. Pilot experiments of grafting 54.12 and 54.6 into Zebrafish indicated that it would require extensive optimization, hence we concentrated on using mouse as the host.

Cryoinjured Tibialis anterior muscles of immunodeficient, NOD/scid/γ chain knockout mice were grafted with 1x106 54.12 FSHD myoblasts, and mice given daily doses of Sunitinib or PBS for 3 weeks, and the muscles removed for analysis. Serial cryosections throughout the length of the muscles were immunostained to measure engraftment efficiency using human-specific antibodies against LAMIN A/C to detect human cells/myonuclei and SPECTRIN to measure myogenic differentiation of donor myoblasts. There was a significant increase in both mean number of LAMIN A/C positive cells in mice treated with Sunitinib compared to controls and number of LAMIN A/C positive cells located within myofibres (myonuclei). Sunitinib treatment also increased the number of SPECTRIN positive myofibres and SPECTRIN-positive/LAMIN A/C-positive myofibres. A more powerful statistical analysis that accounted for biological variation, highlighted the effectiveness of Sunitinib in improving FSHD cell engraftment. In summary, Sunitinib treatment improves the regenerative capacity of FSHD myoblasts by increasing engraftment of cells, and increasing their capacity for myogenic differentiation into SPECTRIN-positive myofibres. This data is now included as Figure 12 in the revised manuscript.

3) In Figure 5, DUX4 OE experiments were used to suggest that Ret is a DUX4 target gene. The authors suggest that Ret is an indirect target that is not directly bound by DUX4 homeodomain. However, the expression of the tMALDUX4-VP16 construct did lead to a increase in Ret expression in Figure 5. In this construct, VP16 is a strong transcriptional activator and should trigger genes near DUX4 binding sites. Thus, it should be considered that Ret could still be a direct DUX4 homeodomain-binding target. The differences between DUX4 and DUX4c are not well defined and the authors should not interpret their results as such. If they do intend to show that Ret is or isn't a direct target of DUX4 DNA-binding, luciferase expression vectors with the proximal promoters of Ret should be used in combination with DUX4 OE to explicitly address this question.

To determine if Ret is a direct target of DUX4, we performed ChIP using C2C12 myoblasts transduced with a truncated DUX4 version that has lower toxicity than full length DUX4, but still contains the DNA binding homeodomains (tMALDUX4-V5). ChIP-qPCR primers were designed around a potential homeodomain binding region approximately 850bp upstream of the transcriptional start site of Ret. There was no significant difference in the enrichment of the tMALDUX4-V5 compared to the IgG control, indicating that DUX4 did not directly bind to this region of the Ret promoter. This data is presented in Figure 6 of the revised manuscript.

Since the dominant negative tMALDUX4-ERD construct also increased Ret expression, and we were unable to show DUX4 binding to the Ret promoter by ChIP, this combined evidence indicated that Ret was not a direct DUX4 target. Thus we felt that obtaining and optimising RET promoter constructs would not be helpful. We have amended the Discussion to include these points.

*4) Furthermore, the dominant negative tMALDUX4-ERD construct also increased Ret expression but was not discussed or mentioned in the text. The use of the Engrailed repressor domain should have shut down the DUX4 pathway, yet Ret is still upregulated. The authors should discuss this phenomenon. Is this the reason why the authors do not think that Ret is a direct target of DUX4?*

That is correct, combined with the ChIP experiment, as detailed in the response above. We have now discussed the implications of these observations in the manuscript.

*5) Caspase 3/7 is not an accurate read out of cell death in the myogenic system. Previous studies have suggested the participation of capspases in the differentiation program. So the proportion of caspase 3/7 staining is affected by changes in differentiation. The authors should perform alternative measurements of cell viability (alamar blue, propidium iodide, etc.).*

We used the Caspase 3/7 release assay in proliferating satellite cell-derived myoblasts in proliferation medium. Furthermore, overexpression of RET proteins does not induce myogenic differentiation in proliferating myoblasts. Thus, any role of Caspase in myogenic differentiation should not interfere with this apoptosis assay to determine if Ret affects apoptosis. However, we have also now measured Annexin V and π in iC2C12-DUX4 myoblasts induced with varying concentrations of Doxycycline and treated with 250ng/ml Sunitinib to measure both apoptosis in response to DUX4 and effects of Sunitinib on DUX4-induced apoptosis. While DUX4 expression increased the proportion of iDUX4-C2C12 myoblasts gated as apoptotic (AV+/PI-, AV-/PI+, AV+/PI+), treatment with Sunitinib had no significant effect. These data have been included as Figure 9.

*6) The authors take a candidate approach to identify RTK inhibitors using phenotypic readouts of cell proliferation and differentiation. However, to demonstrate the pharmacological inhibition of Ret, the authors should provide a read out of Ret inhibition by Sunitinib and other small molecules using a downstream signalling effector.*

We agree with the reviewers’ suggestion. As we have described, constitutively active RET suppressed myogenic differentiation in C2C12 myoblasts and this provides a phenotypic readout of Sunitinib efficacy. To show the pharmacological inhibition of Ret by Sunitinib in myoblasts, we have also assayed phosphorylated ERK and AKT: well characterized downstream effectors of Ret (e.g., Airaksinen and Saarma. Nat Rev Neurosci, 2002. 3(5): p. 383-94). We used iC2C12-DUX4 myoblasts to express DUX4, and measured phosphorylated and total ERK and AKT by quantifying immunoblots. We found that phosphorylated ERK is decreased upon treatment with Sunitinib in DUX4 expressing myoblasts. This is now included as Figure 8.

*7) There seems to be a difference in the function of Ret in wild type myoblasts compared to its function in DUX4 OE or FSHD cells. Knockdown experiments in wild type cells decreased their proliferation, whereas Ret OE led to higher myoblast proliferation. However, in DUX4 OE and FSHD cells Ret inhibition through siRNA knockdown or Sunitinib treatment, proliferation of myoblasts did not change accordingly. The authors should discuss this difference. Are differences in the basal rates of proliferation in DUX4 OE and FSHD cells affected? Do these changes match the predicted change with Ret OE? These differences should be discussed further in the manuscript.*

The Discussion has been amended to include this point.

8) Statistics should be revised throughout the manuscript. Authors are frequently using paired t-test approach (e.g., Figure 1) although there are no paired data. They should consider either simple t-test or even one way ANOVA followed by multiple t-tests. Significance asterisks seem to be random as they are presented for only a subset of bars within graphs although clearly other bars are (appear to be) also significantly different from the controls (e.g., Figure 4).

We have adopted a standardized approach to statistical testing throughout the manuscript.

In cases in which we have made a pairwise comparison between an experimental manipulation and the corresponding control, we have used a t-test assuming a normal distribution of data. When comparing the effects of gene expression or drug application on cultured cells, we have used an unpaired t-test to test significance. When comparing cells derived from the same mouse that have been exposed to different conditions, we have used paired t-tests to test for significance. This is appropriate, provided the pairwise comparisons are made considering cells from the same mouse. For analysis of experiments with multiple parameters to consider, such as origin of cells and effects of a drug, we have used logistical models. These models have allowed us to dissect significant changes caused by single variables in the context of large variation due to other variables. We have better defined which tests are used in the revised manuscript and carefully checked that significant differences are indicated with an asterisk for all t-tests.

[Editors' note: further revisions were requested prior to acceptance, as described below.]

*These concerns can be addressed by modifying some of the claims of the paper, without new experiments.*

*1) A request for clarification at the end of the Discussion: If indeed ERK induces proliferation and inhibits differentiation, ERK inhibition by Sunitinib in DUX4-expressing cells is expected to do the opposite i.e. decrease proliferation besides increasing differentiation. Why is that decreased proliferation presented as an unexpected observation? What are the Authors suggesting as "another mechanism"?*

We have modified the Discussion to remove these ambiguities.

*2) While ChIP qPCR is a method to determine if DUX4 is a direct activator of Ret, the experiment lacks proper controls. There is no positive control to confirm that the ChIP experiment is working. There is also the possibility that the predicted site does not contain the DUX4 binding site.*

*Moreover, there is a rather glancing discussion on why the tMALDUX4-ERD construct also induced Ret expression. The Discussion will benefit from more critical perspectives of the results.*

We have re-evaluated the ChIP experimental data (Figure 6) in light of the recent paper by the Kyba group (Choi et al. (2016) NAR 44, 5161-73) which provides a second ChIP-seq dataset and refines the DUX4 consensus binding site. Since the Ret promoter region does not contain this refined DUX4 consensus site and we could find no ChIP-seq peaks within 15 kb of the Ret TSS, we have now decided to remove our ChIP data. We discuss the implications of these papers for Ret being a direct DUX4 target gene in detail in the revised Discussion. We have also included more clarification about the tMALDUX4-ERD observation.

*3) The identification of pERK1/2 as a readout of Ret activity is also weakly supported. pERK1/2 is not upregulated with DUX4 overexpression, therefore is not likely a Ret target. The decrease in ERK1/2 with Sunitinib treatment yields more questions than answers. It would suggest that Sunitinib is acting on ERK phosphorylation and not Ret activity.*

*While it can be argued that Sunitinib only inhibits ERK when DUX4 is expressed, it could also be due to the co-treatment with Doxycyclin.*

DUX4 up-regulates Ret, and Sunitinib can clearly rescue the differentiation block caused by CA-RET51 in C2C12 myoblasts. Furthermore, ERK1/2 phosphorylation was only inhibited by Sunitinib in DUX4-expressing iC2C12-DUX4 myoblasts. Thus we feel that Sunitinib acting via Ret to inhibit phopsphorylation of ERK is one valid interpretation of the data. However, we have modified the Discussion to accommodate the caveat that Sunitinib could be acting on ERK phosphorylation and not Ret activity. While the observations could also be due to the co-treatment with Doxycyclin, we feel that this is unlikely, as we did not see any effects on ERK phosphorylation with Doxycyclin treatment alone (Figure 8).